# Drosophila-associated bacteria differentially shape the nutritional requirements of their host during juvenile growth

Jessika Consuegra[1][⊛], Théodore Grenier[1][⊛], Patrice Baa-Puyoulet[2], Isabelle Rahioui[2], Houssam Akherraz[1], Hugo Gervais[1], Nicolas Parisot[2], Pedro da Silva[2], Hubert Charles[2], Federica Calevro[2], François Leulier[1]*

**1** Institut de Génomique Fonctionnelle de Lyon, Université de Lyon, École Normale Supérieure de Lyon, Centre National de la Recherche Scientifique, Université Claude Bernard Lyon 1, UMR5242, Lyon, France, **2** Laboratoire Biologie Fonctionnelle, Insectes et Interactions, Université de Lyon, Institut National des Sciences Appliquées, Institut National de Recherche pour l'Agriculture, l'Alimentation et l'Environnement, UMR0203, Villeurbanne, France

⊛ These authors contributed equally to this work.
* francois.leulier@ens-lyon.fr

**Data Availability Statement:** All relevant data are within the paper and in S1 Data file. Metabolic network reconstructions and the resulting BioCyc metabolism databases are available at http://artsymbiocyc.cycadsys.org.

## Abstract

The interplay between nutrition and the microbial communities colonizing the gastrointestinal tract (i.e., gut microbiota) determines juvenile growth trajectory. Nutritional deficiencies trigger developmental delays, and an immature gut microbiota is a hallmark of pathologies related to childhood undernutrition. However, how host-associated bacteria modulate the impact of nutrition on juvenile growth remains elusive. Here, using gnotobiotic Drosophila melanogaster larvae independently associated with Acetobacter pomorum^WJL (Ap^WJL) and Lactobacillus plantarum^NC8 (Lp^NC8), 2 model Drosophila-associated bacteria, we performed a large-scale, systematic nutritional screen based on larval growth in 40 different and precisely controlled nutritional environments. We combined these results with genome-based metabolic network reconstruction to define the biosynthetic capacities of Drosophila germ-free (GF) larvae and its 2 bacterial partners. We first established that Ap^WJL and Lp^NC8 differentially fulfill the nutritional requirements of the ex-GF larvae and parsed such difference down to individual amino acids, vitamins, other micronutrients, and trace metals. We found that Drosophila-associated bacteria not only fortify the host's diet with essential nutrients but, in specific instances, functionally compensate for host auxotrophies by either providing a metabolic intermediate or nutrient derivative to the host or by uptaking, concentrating, and delivering contaminant traces of micronutrients. Our systematic work reveals that beyond the molecular dialogue engaged between the host and its bacterial partners, Drosophila and its associated bacteria establish an integrated nutritional network relying on nutrient provision and utilization.

## Introduction

Nutrition is the major environmental factor that determines to what extent an organism can realize its genetically-encoded growth potential [1]. The attributes of nutrition are defined by the quantity [2], quality [3], and bioavailability [4] of different nutrients in the diet. Nutrients

**Funding:** Research in FL's lab is supported by the "Fondation pour la Recherche Médicale" (Equipe FRM DEQ20180339196) and the Scientific Breakthrough Project from Université de Lyon "Microbehave." Research in PdS and FC's labs are supported by INRA and INSA Lyon. JC is funded by a postdoctoral fellowship from the "Fondation pour la Recherche Médicale" (FRM, SPF20170938612). TG is funded by a PhD fellowship from ENS de Lyon. The funders had no role in study design, data collection and analysis, decision to publish, or preparation of the manuscript.

**Competing interests:** The authors have declared that no competing interests exist.

**Abbreviations:** [ThiS]-COSH, [ThiS]-thiocarboxylate; a.u., arbitrary unit; $Ap^{WJL}$, *Acetobacter pomorum*$^{WJL}$; BCAA, branched-chain amino acid; CFU, colony-forming unit; CR, conventionally raised; CycADS, Cyc Annotation Database System; $D_{50}$, day when 50% of larvae population has entered metamorphosis; DGRP, *Drosophila* Genetic Reference Panel; DT, developmental timing; $EAA^{Fly}$, fly essential amino acid; EC, Enzyme Commission; FAD, Flavin Adenine Dinucleotide; FLYAA, fly exome-matched amino acid ratio; FMN, Flavin mononucleotide; FMOC, 9-fluorenylmethyl chloroformate; GF, germ-free; HD, Holidic Diet; HK, heat-killed; HPLC-MS, High-Performance Liquid Chromatography coupled with Mass Spectrometry; IIS, insulin/insulin-like growth factor signaling; LOD, limit of detection; $Lp^{NC8}$, *Lactobacillus plantarum*$^{NC8}$; MRS, De Man, Rogosa, and Sharpe; *MtnB/C*, genes encoding metallothionein B/C; NAL, nucleic acid and lipid; NCBI, National Center for Biotechnology Information; $NEAA^{Fly}$, fly nonessential amino acid; NMR, Nuclear Magnetic Resonance; $OD_{Max}$, maximal optical density; OPA, ortho-phthalaldehyde; PC, phosphatidylcholine; PE, phosphatidylethanolamine; peg, protein encoding gene; PG, phosphatidylglycerol; PQQ-ADH, pyrroloquinoline-quinone–dependent alcohol dehydrogenase; Tn, transposon; TSP, Trimethylsillyl Propionic Acid; $w^{1118}$, *white*$^{1118}$; WT, wild type; *yw*, *yellow-white*.

are classified as nonessential or essential [3] based on the organism's biosynthetic capacities. Diets deficient in essential nutrients cause important growth and maturation delays or even growth arrest or "stunting", characterized by low height-for-age score [5]. In addition, some nutrients are conditionally essential. These nutrients can be synthesized by the organism but insufficiently under certain metabolically demanding conditions such as juvenile growth. Therefore, these conditionally essential nutrients also need to be retrieved from the diet like the essential ones. Deficient consumption of conditionally essential nutrients can also be detrimental for growth [3].

The intricate relationship between nutrition and growth is modulated by gut microbes. In a classical twin study in humans, Smith and colleagues unequivocally demonstrated that the gut microbiota composition of the juvenile subject suffering from stunting is significantly different from that of the healthy twin. When the fecal microbiota from the discordant twins were transplanted into genetically identical germ-free (GF) mice fed a poor diet, the recipients of the microbiota from the stunted twin performed poorly in terms of growth gain and weight recovery compared to the recipients of the microbiota of the healthy twin [6]. Furthermore, genomic analyses of gut microbiota from children experiencing strong acute malnutrition showed significant under-representation in pathways of amino acid biosynthesis and uptake, carbohydrate utilization, and B-vitamin metabolism [7]. Diets supplemented with nutrients favoring the growth of bacteria enriched in these under-represented pathways increase plasma biomarkers and levels of mediators of growth, bone formation, neurodevelopment, and immune function in children with moderate acute malnutrition [7]. These studies clearly show that microbes strongly impact how organisms respond to changes in their nutritional environment.

Diverse animal models are employed to decipher the physiological, ecological, genetic, and molecular mechanisms underpinning host/microbiota/diet interactions. Among them, *Drosophila melanogaster* is frequently chosen to study the impact of the nutritional environment on growth and development thanks to its short growth period as well as easy and cost-effective rearing conditions. During the juvenile phase of the *Drosophila* life cycle, larvae feed constantly and increase their body mass approximately 200 times until entry into metamorphosis [8]. However, the pace and duration of larval growth can be altered by the nutritional context and the host-associated microbes [9–11]. Like other animals, *Drosophila* live in constant association with microbes, including bacteria and yeast [12]. The impact of the host-associated microbes can be systematically assessed by generating gnotobiotic flies associated with a defined set of bacterial strains [13–15]. Lab-reared flies typically carry bacterial strains from only 4 to 8 species. The microbiota from wild flies are more complex. Nevertheless, they are usually dominated by members of the genera *Acetobacter* and *Lactobacillus* [16–22]. Most bacterial strains from these dominant genera are easy to culture in the lab, and some have even been genetically engineered for functional studies of host–microbe interactions [23–25]. These model bacteria are facultative symbionts that are constantly horizontally acquired [26–28]. Even though recent experimental evidence shows that wild bacterial isolates can persistently colonize the adult crop [22,29], bacteria associated to the larval gut are in fact transient; they constantly shuttle between the larval gut and the food substrate to establish a mutualistic cycle with the host [30,31].

We and others have previously shown that GF larvae raised in poor nutritional conditions show important developmental delays, and association with single model bacterial strains can accelerate *Drosophila* development under these nutritional challenges [20,25]. Specifically, *Acetobacter pomorum*$^{WJL}$ ($Ap^{WJL}$) modulates developmental rate and final body size through the insulin/insulin-like growth factor signaling (IIS) pathway, and its intact acetate production machinery is critical [25]. *Lactobacillus plantarum*$^{WJL}$ or *L. plantarum*$^{NC8}$ ($Lp^{NC8}$) promotes

host juvenile growth and maturation partly through enhanced expression of intestinal peptidases upon sensing bacterial cell wall components by *Drosophila* enterocytes [20,23,32]. Interestingly, the growth-promoting effect of these bacteria is striking under nutritional scarcity, suggesting that besides the molecular dialogue engaged between the bacteria and their host to enhance protein digestion and compensate for reduced dietary macronutrient intake, bacteria-mediated growth promotion on globally scarce diets may also include specific compensation of essential nutrients, as recently reported for thiamin [33]. However, how the presence of such bacteria systematically alters the host's nutritional environment and satisfies the host's nutritional requirements remains unexplored. To do so, we assessed the bacterial contribution to *Drosophila* larval growth in 40 different and strictly controlled nutritional contexts based on chemically defined Holidic Diets (HDs).

HDs comprise a mixture of pure chemical ingredients that satisfy the different physiological requirements of the *Drosophila* host [34,35]. By altering the concentration of each or a combination of ingredients, one can exactly tailor the experiments by generating specific nutrient deficiencies or excess [36]. The first development of HDs supporting the growth of *Drosophila* can be traced back to the 1940s [37], and they were used to assess the direct impact of the nutritional environment on axenic larvae in the 1950s [38,39]. HDs were then used to investigate the links between nutrition and life span [40–43], fecundity [40–42,44], food choice behavior [45,46], nutrient sensing [47], and growth and maturation [33,40–42,48–50]. In this study, we adopted the recently developed fly exome-matched amino acid ratio (FLYAA) HD in which the amino acid concentrations are calculated so that they match the amino acid ratios found in the translated exome of the fly [40]. The FLYAA HD is optimal for both fecundity and life span of adults, and it can efficiently support larval growth, albeit not to the optimal growth and maturation rate obtained with rich oligidic diets [34]. Using this chemically defined HD, we aimed to deconstruct in a systematic manner the microbial contribution to the host's nutritional requirements down to individual nutrients.

To do so, we first needed to establish the biosynthetic capacities of GF larvae and 2 model *Drosophila*-associated bacteria: Ap[WJL] and Lp[NC8] on HD. We further complemented the in vivo study with automated metabolic network reconstruction based on the genome sequences of *D. melanogaster*, Ap[WJL], and Lp[NC8]. In recent years, metabolic approaches based on genome-driven network reconstructions have been applied to predict the potential metabolic dependencies and metabolic exchanges between hosts and associated microbes [51–56]. The mutualistic association between the pea aphid *Acyrthosiphon pisum* and its obligate intracellular symbiont *Buchnera aphidicola* was the first symbiotic association for which genomic information were available on both partners and is a case study for a comprehensive survey of integrated host–symbiont metabolic interactions. In this model, decades of nutritional experiments using HDs and aposymbiotic aphids were reinterpreted in the light of newly available genomic data, thus changing the traditional paradigm that proposed a clear separation between the pathways of the host and its symbionts and revealing a particularly integrated metabolic network that is the result of the long coevolution of the insect with its obligate endosymbionts [57,58]. This example shows how important it is to integrate theoretical and experimental approaches to model metabolic pathways of symbiotic partners and properly dissect the functioning of their associations.

Here, we report that association of GF larvae with Ap[WJL] or Lp[NC8] modifies the nutritional requirements of ex-GF larvae in a specific manner for each bacterium. We show that Ap[WJL] and Lp[NC8] not only modify the nutritional environment of their host by fortifying diets with essential nutrients, they functionally compensate host auxotrophies despite not synthetizing the missing nutrient, probably by either providing a nutrient derivative to the host or by uptaking, concentrating, and delivering contaminant traces of the missing micronutrient.

## Results and discussion

Metabolic network reconstruction of the host (*D. melanogaster*) and its associated bacteria, Ap$^{WJL}$ and Lp$^{NC8}$, was automatically generated using the Cyc Annotation Database System (CycADS) pipeline [59]. The resulting BioCyc metabolism databases are available at http://artsymbiocyc.cycadsys.org for annotation and analysis purposes. We generated the enriched functional annotations of all the predicted proteins from the complete genomes of *D. melanogaster* (*Drosophila*, RefSeq GCF_000001215.4 release 6), *A. pomorum* strain DM001 (Ap$^{WJL}$, accession National Center for Biotechnology Information [NCBI] Bioproject PRJNA60787), and *L. plantarum* subsp. *plantarum* NC8 (Lp$^{NC8}$, NCBI Bioproject PRJNA67175). From the genomic analyses, we inferred all pathways allowing production of the organic compounds that are present in the exome-based FLYAA HD developed by Piper and colleagues [40]: fly essential and nonessential amino acids (EAAs$^{Fly}$ ($n = 10$) and NEAAs$^{Fly}$ ($n = 10$)), B-vitamins ($n = 7$), cholesterol ($n = 1$), and nucleic acids and lipid precursors (NALs, $n = 4$).

### *D. melanogaster* biosynthetic capabilities inferred from genome-based metabolic network reconstruction

Although a BioCyc metabolic reconstruction of *D. melanogaster* is already publicly available (https://biocyc.org/FLY), we constructed an improved BioCyc database using a recent genome version and annotation [59]. This metabolic reconstruction identified 22,189 protein-encoding genes, including 5,061 enzymes and 156 transporters associated with 1,610 compounds assembled in a network of 331 pathways (versus the 227 pathways found in BioCyc). Like other metazoans, *Drosophila* possesses the gene repertoire to produce all the NEAAs$^{Fly}$ but is unable to produce the EAAs$^{Fly}$ (Fig 1A and S1 Table). *Drosophila* can also produce myoinositol, inosine, and uridine but is unable to synthesize vitamins from simple precursors (Fig 1B and S2 Table).

### Ap$^{WJL}$ biosynthetic capabilities inferred from genome-based metabolic network reconstruction

According to our metabolic reconstruction, the Ap$^{WJL}$ genome comprises 4,268 protein-encoding genes including 1,326 enzymes and 46 transporters associated with 1,306 compounds assembled in a network of 313 pathways. Ap$^{WJL}$ is a complete autotroph for all amino acids and possesses the genetic potential to produce the DNA bases inosine and uridine and 5 of the 7 vitamins present in the HD: biotin, folate, pantothenate, riboflavin, and thiamine (Fig 1A and 1B and S1 and S2 Tables). The first 2 steps of the nicotinate pathway (Enzyme Commission [EC] number 1.4.3.16 and 2.5.1.72) seem lacking in Ap$^{WJL}$. However, 3 candidate proteins (protein encoding genes [pegs].1228, 1229, and 1231) encode the succinate dehydrogenase enzymatic activity (EC 1.3.5.1). This enzyme can alternatively use oxygen or fumarate as an O-donor, depending on aerobic or anaerobic living conditions. Hence, this enzyme can switch between its aerobic condition activity (EC 1.3.5.1) towards its anaerobic condition activity (EC 1.4.3.16) using fumarate as a substrate and producing imminoaspartate. Hence, assuming that one of these genes can produce the activity at a sufficient rate in aerobic conditions in Ap$^{WJL}$, then the bacteria would be able to produce NAD$^+$ and NADP$^+$ from Asp (Fig 1B and S2 Table). The biosynthesis of pyridoxine is almost complete in Ap$^{WJL}$. Although we were not able to detect specific activities for the first 2 steps of the pathway, we propose (see below) that the bacteria have the capability to produce vitamin intermediates using enzymes with very close activities (S2 Table). Note that pyridoxine is reported as nonessential for acetic acid bacteria [60]. In summary, Ap$^{WJL}$ genome analysis predicts that it is able to synthesize all

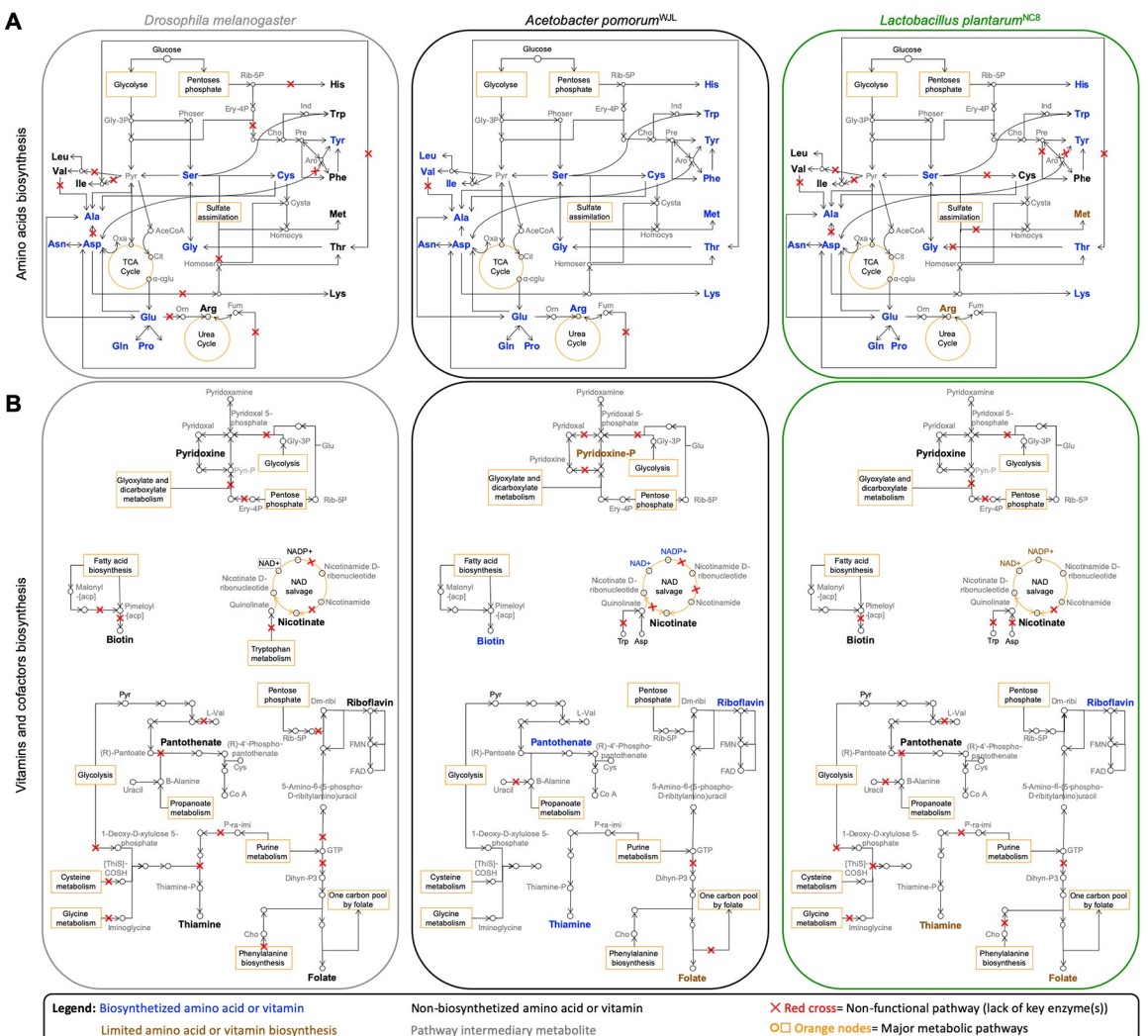

**Fig 1. Expert automated genome annotation and metabolic network reconstruction of *Drosophila*, Ap^WJL, and Lp^NC8.** (A) Amino acid biosynthetic pathways. (B) Vitamins and cofactors biosynthetic pathways. Left panels, *D. melanogaster*. Central panels, Ap^WJL. Right panels, Lp^NC8. Color code: blue, biosynthesized amino acids or vitamins; brown, limited amino acid or vitamin biosynthesis (biosynthesis of the metabolite may be possible, but it is limited and/or requires secondary metabolic pathways); black, nonbiosynthesized amino acids or vitamins; gray, pathway intermediary metabolites. Red cross: nonfunctional pathway (lack of key enzyme[s]). Orange nods, major metabolic pathways. α-cglu, α-keto-glutarate; AceCoA, Acetyl-CoA; Ant, Antranilate; Ap^WJL, *A. pomorum*^WJL; Aro, Arogenate; Cho, chorismate; Cit, Citrate; Cysta, Cystathionine; Dihyn-P3, 7,8-Dihydroneopterin-3′-P3; Dm-ribi, 6,7-Dimethyl-8-ribityllumazine; Ery-4P, Erythrose-4P; FAD, Flavin Adenine Dinucleotide; FMN, Flavin mononucleotide; Fum, Fumarate; Glc, Glucose; Gly-3P, Glycerate-3P; Homocys, Homocysteine; Homoser, Homoserine; Ind, Indole; Lp^NC8, *L. plantarum*^NC8; Orn, Ornithine; Oxa, Oxaloacetate; P-ra-imi, 1-(5′-Phospho-ribosyl)-5-aminoimidazole; Phoser, Phosphoserine; Pre, Prephenate; Pyn-P, Pyridoxine phosphate; Pyr, Pyruvate; Rib-5P, Ribose-5P; TCA, Tricarboxylic acid Cycle; [ThiS]-COSH, [ThiS]-thiocarboxylate.

amino acids, DNA bases, and the 7 B-vitamins (biotin, folate, pantothenate, riboflavin, thiamine, and intermediates of nicotinate and pyridoxine) present in HDs. However, we found no genomic support for the synthesis of choline and myoinositol in the Ap^WJL genome.

## Lp^NC8 biosynthetic capabilities inferred from genome-based metabolic network reconstruction

Metabolic reconstruction from the Lp^NC8 genome generated a database that includes 2,868 protein-encoding genes, consisting of 973 enzymes and 74 transporters associated with 1,154

compounds, all assembled in a network of 246 metabolic pathways. From a genomic perspective (Fig 1A and S1 and S2 Tables), Lp$^{NC8}$ is able to produce most amino acids from glucose or inner precursors with the exception of Phe, sulfur-containing amino acids (Cys, Met), and branched-chain amino acids (BCAAs; Ile, Leu, Val). Arg is known to be limiting [61] or essential to certain *L. plantarum* strains [62,63], yet the Lp$^{NC8}$ genome encodes a complete Arg biosynthesis pathway. A manual curation of the pathway showed that the Lp$^{NC8}$'s *argCJBDF* operon should be functional because it does not contain stop codons, frameshifts, or deletions. Lp$^{NC8}$ may produce Ala and Asp only using secondary metabolic routes (S1 Table). Therefore, Lp$^{NC8}$ is expected to acquire these amino acids from the diet or to have an altered growth when they are absent from the diet. Similarly, biosynthesis of Thr is directly linked to Asp and Cys and is probably very limited in Lp$^{NC8}$.

Regarding vitamins and bases biosynthesis, Lp$^{NC8}$ is able to produce folate, riboflavin, and thiamine (through the pyrimidine salvage pathway [2.1.7.49]), as well as all DNA bases including uridine and inosine (Fig 1B and S2 Table). Lp$^{NC8}$ is not able to synthesize biotin, pyridoxine, pantothenate, choline, and myoinositol. Based on our genomic analysis, Lp$^{NC8}$ is not able to achieve the entire nicotinate biosynthetic pathway from Asp nor from Trp, as described in eukaryotes and in some bacteria [64]; even if the first step of the pathway could possibly be accomplished by the succinate dehydrogenase, as described above for Ap$^{WJL}$, the other 2 enzymes of the initial part of the pathway are missing (Fig 1B and S2 Table).

Collectively, our metabolic networks reconstruction shows that *Drosophila* and its associated bacteria have differential biosynthetic capacities. Indeed, some of the complete biosynthetic pathways are only present in one organism, while others are present in 2 or all 3 partners (Fig 2). In addition, we did not detect incomplete biosynthetic pathways potentially complemented between the host and its associated bacteria (Figs 1 and 2), as previously observed for obligate mutualistic partners [57,58].

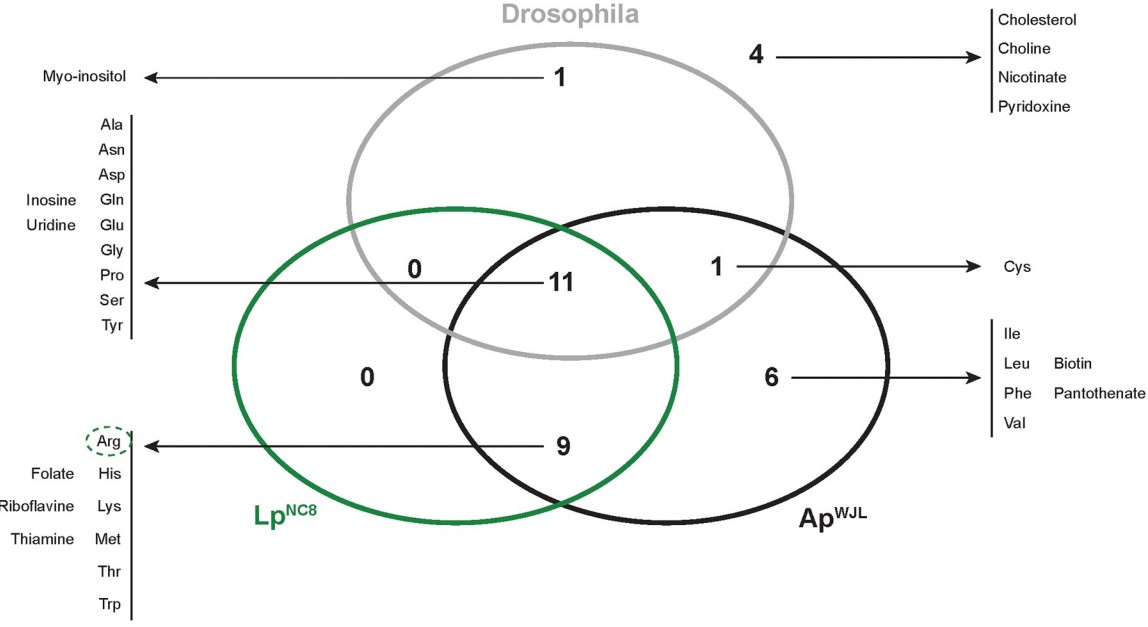

**Fig 2. *Drosophila*, Ap$^{WJL}$, and Lp$^{NC8}$ have differential biosynthetic capacities of nutrients contained in the HD.** Venn diagram represents the number of nutrients present in the FLYAA HD that can be synthesized by each organism. The list of corresponding metabolites is provided. Dotted circles: biosynthesis of this metabolite by Lp$^{NC8}$ (green) may be possible but might be limiting. Ap$^{WJL}$, *A. pomorum*$^{WJL}$; FLYAA, fly exome-matched amino acid ratio; HD, Holidic Diet; Lp$^{NC8}$, *L. plantarum*$^{NC8}$.

## Experimental validation of *Drosophila*-associated bacteria auxotrophies using HDs

In order to experimentally test the metabolic potential of *Drosophila* and its associated bacteria predicted by our automated genome annotations and subsequent metabolic pathway reconstructions (see above), we adopted the exome-based FLYAA HD [40]. We systematically removed a single component at a time to generate 39 different fly nutritional substrates (henceforth named HDΔX, X being the nutrient omitted), plus one complete HD medium. This medium can also be prepared in a liquid version by omitting agar and cholesterol from the recipe. Liquid HDs can then be used to assess bacterial growth in 96-well plates, increasing the experimental throughput.

We first assessed $Ap^{WJL}$ and $Lp^{NC8}$ growth in each of the 40 different liquid HDs for 72 h, using maximal optical density ($OD_{Max}$) as a readout (Fig 3A and S3 Table). In the complete HD, both $Ap^{WJL}$ and $Lp^{NC8}$ grow well (Fig 3A, first line). On the deficient media, $Ap^{WJL}$ can grow in HDΔSucrose, presumably using acetate from the acetate buffer as a carbon source. Also, its growth is not altered in the absence of any of $EAAs^{Fly}$, vitamins, or NALs. However, while $Ap^{WJL}$ growth is not impacted by the lack of most $NEAAs^{Fly}$, it grows poorly in

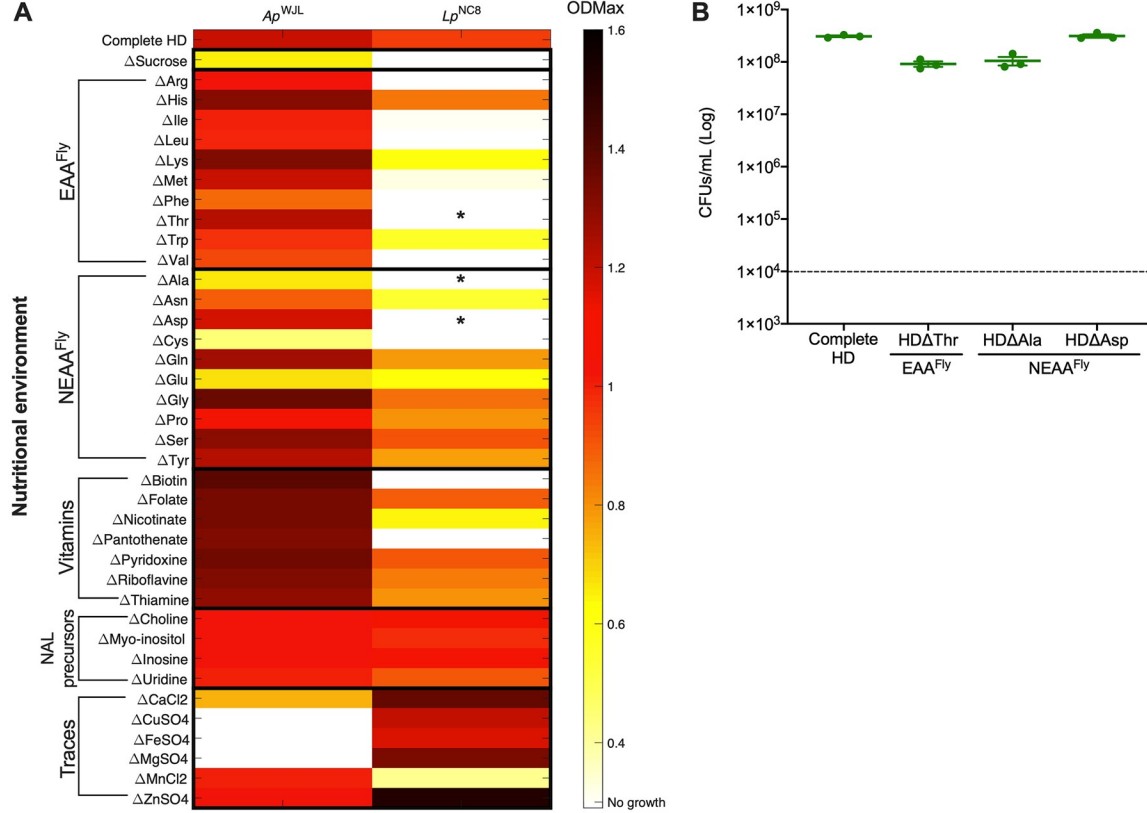

**Fig 3. $Ap^{WJL}$ and $Lp^{NC8}$ auxotrophies detected in liquid fly HD.** (A) Heat map representing the mean $OD_{Max}$ reached by $Ap^{WJL}$ or $Lp^{NC8}$ after 72 h of culture. Each line shows growth in a different version of the liquid HD: complete HD (first line) or HD lacking nutrient X (ΔX, lines below). Cultures were made in 96-well plates under agitation. Asterisks (*) pinpoint contradictions with our metabolic pathway automated annotations, which are explained in panel B. (B) Growth of $Lp^{NC8}$ in 4 versions of liquid HD: complete HD, HDΔThr, HDΔAla, and HDΔAsp in static conditions. Plot shows means with standard error based on 3 replicates by assay. Each dot represents an independent replicate. The dashed line represents the level of inoculation at $t = 0$ h ($10^4$ CFUs per mL). $Ap^{WJL}$, *A. pomorum*$^{WJL}$; CFU, colony-forming unit; $EAA^{Fly}$, fly essential amino acid; HD, Holidic Diet; $Lp^{NC8}$, *L. plantarum*$^{NC8}$; NALs, nucleic acids and lipids; $NEAA^{FLY}$, fly nonessential amino acid; $OD_{Max}$, maximal optical density.

HDΔAla, HDΔCys, and HDΔGlu. In addition, Ap$^{WJL}$ fails to grow in HDΔCu, HDΔFe, and HDΔMg (Fig 3A, first column and S3 Table). The broad growth capacity of Ap$^{WJL}$ in HDs correlates well with the wide range of environmental niches the genus *Acetobacter* can colonize. *Acetobacter* species are found in sugar-rich niches such as flowers and fruits but also in poorer niches such as soil and water, where they need to synthesize all the nutrients required for their own growth [65]. These findings corroborate our genome-based predictions (Fig 1). Furthermore, the genome-based metabolic pathway reconstruction predicted that Ap$^{WJL}$ would not be able to synthesize choline and myoinositol; however, we observed that Ap$^{WJL}$ grows in their absence. Choline is an important precursor of phosphatidylcholine (PC), which is a major component of *Acetobacter* membranes and plays an important role in conferring acetic acid tolerance. Despite its importance, PC is not essential for *Acetobacter* growth. Indeed, mutants precluding PC synthesis show a shift towards increased membrane content of phosphatidylethanolamine (PE) and phosphatidylglycerol (PG) and do not show any growth defects in standard medium [66]. Similarly, Ap$^{WJL}$ likely does not need myoinositol for its growth because inositol compounds are absent from the membrane of most bacteria [67]. Regarding nicotinate and pyridoxine, the biosynthesis pathways of these 2 vitamins are only partial and do not support the production of the final molecules (Fig 1B and S2 Table); however, intermediates such as pyridoxine phosphate, pyridoxal-5-phosphate, and pyridoxamine or nicotinate-D-ribonucleotide, NAD$^+$, and NADP$^+$ may be synthesized and would support bacterial growth in nicotinate- or pyridoxine-depleted diets. Interestingly, Ap$^{WJL}$ growth was only precluded in the absence of some metal ions: Cu, Fe, and Mg. Metal ions are important cofactors required for enzymatic activities [68]. Specifically, in acetic acid bacteria, Cu is an important cofactor of the energy-producing cytochromes of the respiratory chain [69], making it essential for Ap$^{WJL}$ growth.

We detected far more nutritional auxotrophies for Lp$^{NC8}$ on HDs (Fig 3A, second column and S3 Table). Lp$^{NC8}$ fails to grow in HDΔSucrose because sucrose is the only suitable carbon source for this strain in the liquid HD. Also, Lp$^{NC8}$ growth is precluded in the absence of 9 amino acids, including 6 EAAs$^{Fly}$ (Arg, Ile, Leu, Phe, Thr, Val) and 3 NEAAs$^{Fly}$ (Ala, Asp, Cys). It also grows poorly in media lacking the EAAs$^{Fly}$ Lys, Met, and Trp and the NEAAs$^{Fly}$ Asn and Glu. Moreover, Lp$^{NC8}$ does not grow in HDΔBiotin and HDΔPantothenate. However, it slightly grows in absence of nicotinate, despite the prediction from our genome-based metabolic pathway reconstruction that nicotinate could not be produced (Fig 1B and S2 Table). Finally, Lp$^{NC8}$ growth is not affected by the lack of any NALs and even increased in the absence of certain metal ions such as Ca, Cu, Mg, and Zn. In contrast, Lp$^{NC8}$ growth is significantly reduced in HDΔMn. These relatively elevated nutritional requirements of Lp$^{NC8}$ were expected because *L. plantarum* is a species adapted to nutrient-rich environments [70]. Hence, many *L. plantarum* strains have lost the capacity to synthesize various nutrients that can easily be found in their natural habitats [70,71]. The inability of *L. plantarum* to synthesize important nutrients such as BCAAs (Ile, Leu, and Val) or the B-vitamin pantothenate was previously identified by both genome analyses [62] and growth studies in chemically defined minimal media [61,72,73]. Moreover, it is known that *L. plantarum* needs Mn to resist oxidative stress [74], which explains its poor growth in HDΔMn.

Our experimental data only partially correlate with the results of the genome-based predictions. Predicted auxotrophies for Ile, Leu, Val, Phe, Cys, pantothenate, and biotin were confirmed in vivo. The identified Arg auxotrophy was not surprising because, as mentioned above, Arg is often described as essential to *L. plantarum* in high-metabolic–demanding conditions even though all the genes necessary for Arg biosynthesis are present. However, auxotrophies of Lp$^{NC8}$ to Thr, Ala, and Asp were not expected (Fig 3A, denoted by "*"), even though these amino acids were predicted to be limiting (see above). As mentioned previously, bacterial

growth in liquid HDs was assessed in 96-well plates using a microplate reader (see Materials and methods). Every cycle includes an agitation step to homogenize the solution to improve OD reading accuracy. This agitation step may oxygenate the media and thus negatively affects Lp$^{NC8}$ growth in suboptimal nutritional conditions because *L. plantarum* strains are aerotolerant, but optimal growth is achieved under microaerophilic or anaerobic conditions [75]. To challenge these unexpected auxotrophies, we assessed Lp$^{NC8}$ growth in liquid HDΔThr, HDΔAla, and HDΔAsp in 15-mL closed falcon tubes without agitation. After 72 h of incubation, we determined colony-forming unit (CFU) counts in each media (Fig 3B). As predicted by our genomic analyses, Lp$^{NC8}$ was now able to grow in each of the 3 deficient media in static conditions to the same extent as in the complete HD (Fig 3B). Therefore, Lp$^{NC8}$ auxotrophies observed for Thr, Ala, and Asp in 96-well plates are likely due to excessive oxygenation. This could also explain the poor growth of Lp$^{NC8}$ in the absence of the EAAs$^{Fly}$ Lys, Met, and Trp and the NEAAs$^{Fly}$ Asn and Glu.

Surprisingly, the ability of Lp$^{NC8}$ to grow in HDΔCholine, HDΔMyoinositol, HDΔNicotinate, and HDΔPyridoxine does not correlate with our metabolic predictions. As for Ap$^{WJL}$ (see above), Lp$^{NC8}$ growth probably does not require choline or myoinositol. A previous study quantified choline and inositol compounds in *L. plantarum* cell extracts and found them to be extremely low and therefore most likely due to contaminations from the medium rather than components of *L. plantarum* biomass [76]. Pyridoxine is a precursor of pyridoxal-5-phosphate, a cofactor necessary for amino acid converting reactions. Teusink and colleagues [62] showed that *L. plantarum*$^{WCSF1}$ requires exogenous sources of pyridoxine only in a minimal medium lacking amino acids. Because HDΔPyridoxine contains all amino acids, it is likely that pyridoxine is not essential for Lp$^{NC8}$ growth in these conditions. Finally, the capacity of Lp$^{NC8}$ to grow in HDΔNicotinate could be related to the presence of alternative pathways to nicotinate intermediate biosynthesis (Fig 1B and S2 Table). Indeed, this possibility has been previously reported in the genus *Lactobacillus* [71], which would explain the capacity to grow in absence of exogenous nicotinate.

Altogether, the complete HD is a suitable nutritional environment that allows the 2 model *Drosophila*-associated bacteria, Ap$^{WJL}$ and Lp$^{NC8}$, to grow. Growth capacities in deficient media vary from one bacterium to another and are dictated by their individual genetic repertoires.

## GF larvae exhibit 22 auxotrophies while developing on FLYAA HDs

We next sought to establish the nutritional requirements of GF larvae by assessing larval developmental timing (DT) in the complete HD and in each of the 39 deficient HDs (larvae were reared from eggs until pupae on the HDs; see Materials and methods). DT is expressed as D$_{50}$, which represents the day when 50% of the larvae population has entered metamorphosis in a specific nutritional condition. In agreement with previous studies [38,39], GF larvae fail to develop in all HDΔEAAs$^{Fly}$, all HDΔVitamins, HDΔCholine, HDΔCholesterol, HDΔZn, and HDΔMg (Fig 4A, first column). Over 60 years ago, Sang and colleagues reported that Zn was dispensable for GF larval development [38]. We suspect that the casein in the medium used in Sang and colleagues inadvertently provided trace amount of Zn, which could account for the discrepancy between our observation and that of Sang and colleagues. Also in accordance with previous studies [38,39,50], GF larvae were able to reach pupariation in HDΔNEAAs$^{Fly}$ (ΔAla, ΔCys, ΔGln, ΔGlu, ΔGly, ΔPro), HDΔUridine, HDΔMyoinositol, and HDΔMn at the same rate as on a complete HD (Fig 4A first column, S4 Table). The absence of sucrose, Tyr, inosine, Ca, Cu, and Fe did not prevent pupae emergence but increased the duration of larval development very significantly (Fig 4A first column, S4 Table). Surprisingly, GF larvae were able to

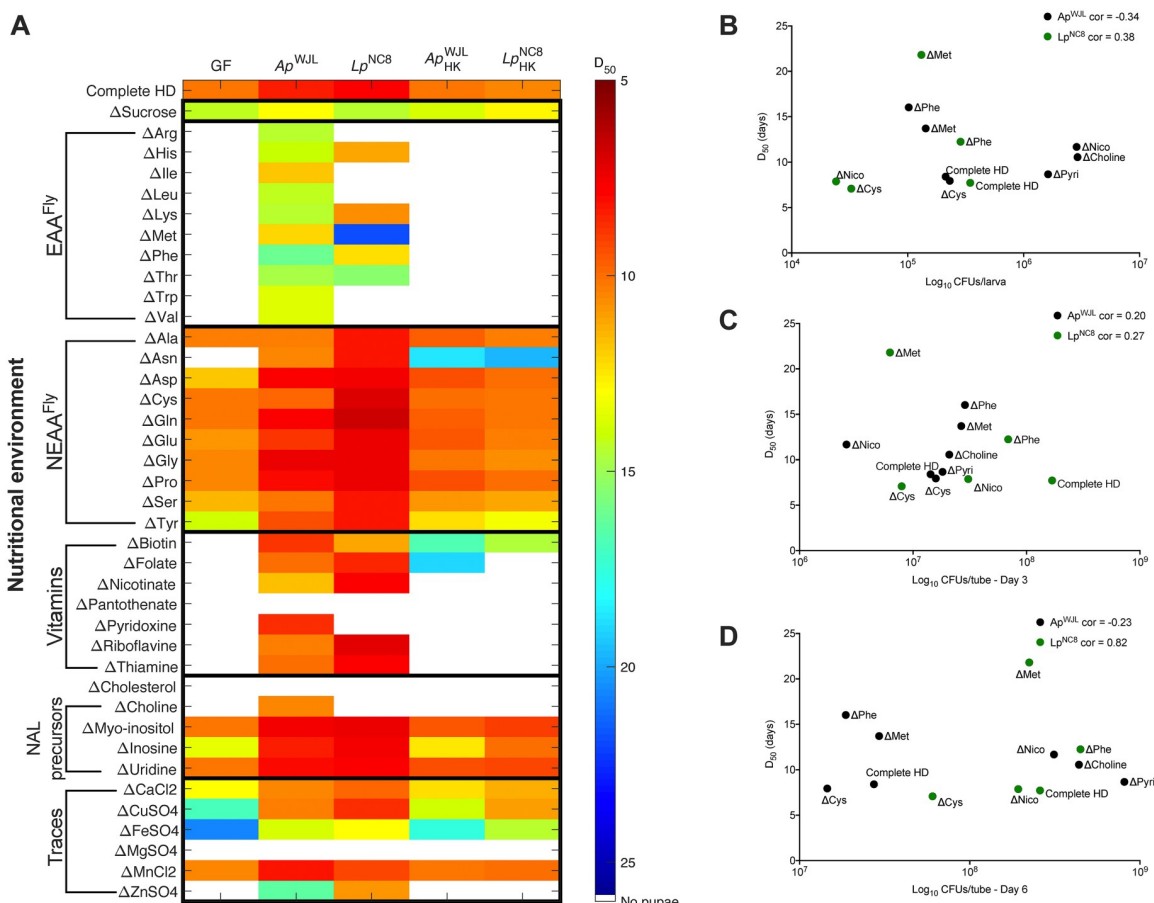

**Fig 4. Ap^WJL and Lp^NC8 can differentially fulfill their host's nutritional requirements in HDs.** (A) Heat map representing the mean $D_{50}$ of GF larvae (first column) and larvae associated with Ap^WJL, Lp^NC8, Ap^WJL_HK, and Lp^NC8_HK (columns 2, 3, 4, 5 respectively). Each line shows $D_{50}$ in a different version of HD: complete HD (first line) or HDs lacking nutrient X (ΔX, lines below). White means larvae did not reach pupariation in these conditions. Means, standard errors of the mean and statistical tests (Dunn test of multiple comparisons) are detailed in S4 Table. (B–D) Absence of correlation between time of development and quantity of bacteria. *Y* axis shows $D_{50}$, and *X* axis shows quantity of bacteria (Log_10 CFUs) in the larval gut (B), in the diet in presence of larvae 3 days after inoculation (C), and in the diet in presence of larvae 6 days after inoculation (D). Each dot shows a different condition. Complete HD: on complete HD. ΔX: on HDs lacking nutrient X. Black dots: in monoassociation with Ap^WJL, green dots: in monoassociation with Lp^NC8. For each bacterium, we tested Pearson's product–moment correlation between $D_{50}$ and quantity of bacteria. Ap^WJL, *A. pomorum*^WJL; CFU, colony-forming unit; cor, Pearson correlation coefficient for each bacterium; $D_{50}$, day when 50% of larvae population has entered metamorphosis; EAA^Fly, fly essential amino acid; GF, germ-free; HD, Holidic Diet; HK, heat-killed; Lp^NC8, *L. plantarum*^NC8; NALs, nucleic acids and lipids; NEAA^Fly, fly nonessential amino acid.

reach pupariation, albeit late, in HDΔSucrose. Indeed, all the HDs developed to date include carbohydrates (either sucrose or fructose) as a carbon source [34]. Larval development in the absence of carbohydrates suggests that GF larvae may use other components of the HD such as amino acids as carbon source. In summary, GF *yellow-white* (*yw*) larvae show 22 auxotrophies while developing on sterile HDs.

Our observations correlate well with our genome-based predictions of the metabolic capabilities of the 3 partners (Fig 1) with one exception: GF larvae did not reach pupariation in HDΔAsn. This result was surprising because Asn is described as an NEAA in *Drosophila* and other animals [77]. To test whether Asn auxotrophy was specific to the *yw* fly line used in our lab, we assessed larval DT in 2 other *D. melanogaster* reference lines, the *Drosophila* Genetic Reference Panel (DGRP) line DGRP_25210 [78] and *white*^1118 (*w*^1118). Unlike *yw*, both *w*^1118

and DGRP_*25210* larvae were able to develop in GF conditions in HDΔAsn, albeit with a severe developmental delay (Fig 5A). Therefore, the complete Asn auxotrophy seen with our *yw* strain is an exception rather than a rule, an observation that correlates with our metabolic pathway reconstruction that was based on the genome sequence of the *D. melanogaster* reference genome strain (Bloomington stock #2057). We next sequenced the coding region of the enzyme AsnS, which converts Asp to Asn in *yw* flies, and did not detect any nonsynonymous mutation (S1 Fig). Further studies may thus be required to determine the origin of the Asn auxotrophy in our *yw* line on HD. However, these results indicate that Asn is not an EAA per se but remains a limiting NEAA, an observation that also applies to Tyr.

## Bacterial cell wall sensing contributes to Lp^NC8^-mediated larval growth promotion in complete chemically defined diets

We then investigated whether and how the association with bacteria affects the nutritional requirements of GF larvae during juvenile growth and maturation. To this end, we monoassociated GF embryos with Ap^WJL^ or Lp^NC8^ and measured $D_{50}$ and egg-to-pupa survival in complete and deficient HDs (Fig 4A, second and third columns, respectively, and S4 and S5 Tables). On a complete HD, monoassociation with either Ap^WJL^ or Lp^NC8^ accelerated larval DT with a mean $D_{50}$ of 8.4 and 7.7 days, respectively, whereas GF mean $D_{50}$ is 10.1 days (Fig 4A, first line). These growth-promoting effects upon monoassociation with either Ap^WJL^ or Lp^NC8^ have been previously reported on complex diets, and insights on the underlying molecular mechanisms were provided [20,25]. Shin and colleagues showed that when the associated larvae grow on a low-casamino–acid semioligidic diet, the pyrroloquinoline-quinone–dependent alcohol dehydrogenase (PQQ-ADH) activity of Ap^WJL^ modulates the developmental rate and body size through IIS. PQQ-ADH transposon (Tn) disruption in the Ap^WJL^::Tn*pqq* mutant severely reduces acetic acid production, which has been proposed to alter the

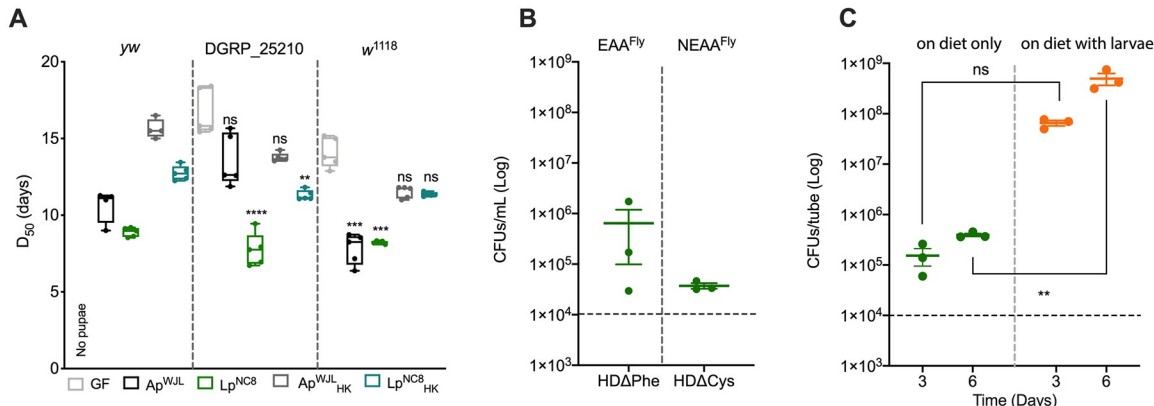

**Fig 5. Evaluation of HDΔAsn, HDΔPhe, and HDΔCys contexts.** (A) $D_{50}$ of *yw*, DGRP_*25210*, and *w*^1118^ larvae on HDΔAsn. Boxplots show minimum, maximum, and median. Each dot shows an independent replicate. GF *yw* larvae did not reach pupariation. For the other 2 lines, we performed a Kruskal–Wallis test followed by post hoc Dunn tests to compare each gnotobiotic condition to GF. **$p$-value < 0.005, ***$p$-value < 0.0005, ****$p$-value < 0.0001. (B) Growth of Lp^NC8^ in liquid HDΔPhe and liquid HDΔCys, in static conditions, 3 days after inoculation. Plot shows mean with standard error. Each dot shows an independent replicate. The dashed line represents the level of inoculation at $t = 0$ h ($10^4$ CFUs per mL). (C) Growth of Lp^NC8^ on solid HDΔCys, in absence and in presence of larvae, 3 days and 6 days after inoculation. Plot shows mean with standard error. Each dot represents an independent replicate. The dashed line represents the level of inoculation at $t = 0$ h ($10^4$ CFUs per tube). We performed two-way ANOVA followed by post hoc Sidak test. **$p$-value < 0.005. Ap^WJL^, *A. pomorum*^WJL^; CFU, colony-forming unit; DGRP, XXX; $D_{50}$, day when 50% of larvae population has entered metamorphosis; EAA^Fly^, fly essential amino acid; GF, germ-free; HD, Holidic Diet; HK, heat-killed; Lp^NC8^, *L. plantarum*^NC8^; NEAA^Fly^, fly nonessential amino acid; ns, nonsignificant; *yw*, XXX.

regulation of developmental and metabolic homeostasis upon monoassociation [25]. Lp^NC8 promotes host juvenile growth and maturation on a low-yeast–based oligidic diet, partly through enhanced expression of intestinal peptidases upon sensing of bacterial cell walls components by *Drosophila* enterocytes [20,23]. Deletion of the *dlt* operon, which encodes the molecular machinery involved in the D-alanylation of teichoic acids, leads to bacterial cell wall alteration with a complete loss of D-alanylation of teichoic acids, and consequently, cell walls purified from the Lp^NC8Δ*dlt*op mutant trigger a reduced expression of peptidases in enterocytes [23]. Therefore, we first probed the importance of these molecular mechanisms on bacteria-mediated larval growth promotion on a complete HD. To this end, we tested in our HD setting the associations with the loss of function mutants Ap^WJL::Tn*pqq* [25] and Lp^NC8Δ*dlt*op [23]. In a complete HD, only the Lp^NC8Δ*dlt*op mutant failed to support larval growth, reminiscent of the previous observation on the low-yeast oligidic diets (S2 Fig). Surprisingly, in complete HD, the Ap^WJL::Tn*pqq* mutant actually triggered an enhanced growth promotion as compared to its wild-type (WT) reference strain. Shin and colleagues reported that Ap^WJL::Tn*pqq*-associated larvae experienced growth delay, which can be rescued by acetic acid provision [25]. Therefore, the acetic-acid–based buffer in the HD may explain why Ap^WJL::Tn*pqq* no longer behaves as a loss-of-function mutant in this setting; however, how it actually surpasses the WT strain on a complete HD remains elusive. Collectively, these results establish that sensing bacterial cell walls containing D-alanylated teichoic acids is also an important feature of the intrinsic growth-promoting ability of Lp^NC8 in a complete chemically defined HD. Thus, the previously reported molecular sensing mechanism that mediates the growth-promoting effect of Lp^NC8 during chronic undernutrition is also at play in synthetic diets.

## Association with Ap^WJL fulfills 19 of the 22 nutrient requirements of GF larvae

Association with Ap^WJL sustained larval development (albeit to different degrees) in the absence of 19 out of 22 GF larvae essential nutrients (Fig 4A, second column). Ap^WJL-associated larvae reached pupariation in the absence of each EAA^Fly (though their development was slower than on complete HD), Asn, vitamins, choline, and Zn. Association with Ap^WJL also rescued the developmental delay observed in GF larvae in HDΔTyr, HDΔinosine, HDΔCu, and HDΔFe. The only nutritional requirements of GF larvae that were not fulfilled by Ap^WJL were cholesterol, pantothenate, and Mg.

## Association with Lp^NC8 fulfills 12 of the 22 nutrient requirements of GF larvae

Compared to Ap^WJL, monoassociation with Lp^NC8 compensated for a reduced number of the GF larvae nutritional deficiencies (12 out of 22; Fig 4A, third column). Lp^NC8-associated larvae reached pupariation in the absence of some EAAs^Fly (HDΔHis, HDΔLys, HDΔMet, HDΔPhe, HDΔThr), Asn, certain vitamins (HDΔBiotin, HDΔFolate, HDΔNicotinate, HDΔRiboflavin, HDΔThiamine), and Zn. Moreover, Lp^NC8 rescued the developmental delay observed in GF larvae on HDΔTyr, HDΔinosine, HDΔCu, and HDΔFe.

## Bacteria need to be metabolically active in order to fulfill larval nutritional requirements

Bacteria were grown in rich medium before association with larvae (see Materials and methods). Therefore, they might have accumulated nutrients that could be used later by the larvae to fulfill their nutritional requirements. To test for the nutritional input brought by the initial

bacterial inoculum, we associated GF larvae with 10× heat-killed (HK) bacteria (mimicking the maximal bacterial biomass found in the diet during the experiment, S3B Fig) and measured $D_{50}$ in complete and deficient HDs (Fig 4A, fourth and fifth columns). In most cases, the $D_{50}$ of larvae in HK and GF conditions was similar. Therefore, bacteria need to be metabolically active to fulfill the larval nutritional requirements on HDs. However, we found some exceptions. In HDΔAsn, HDΔBiotin, HDΔFolate, HDΔCu, and HDΔFe, the addition of HK bacteria allowed the larvae to develop, though not as fast as in association with living bacteria. These results suggest that larvae only require a very small amount of these nutrients, which can be sufficiently derived from the inert bacterial inoculum.

On a low-protein oligidic diet, larval growth promotion by bacteria correlates positively with the quantity of microbes [79]. We wondered whether the differences that we observed in growth-promotion efficiency were due to differences in bacterial loads. Thus, we tested the correlation between bacterial loads and benefit to host growth in 3 contexts: (1) conditions in which both bacteria are beneficial to their host, complete HD; (2) conditions in which each bacterium differently impacts the host, HDΔMet, HDΔPhe, and HDΔNicotinate; and (3) conditions in which only one bacterium compensates for the lack of a nutrient, HDΔCys, HDΔPyr, and HDΔCholine. We found no correlation between bacterial load in the larval gut (Fig 4B and S3A Fig) or in the diet (Fig 4B and 4C and S3B Fig) and the ability of the bacteria to impact host DT on the tested diets. These results reinforce the notion that, in our experimental settings, *Drosophila*-associated bacteria are biologically active partners, and their load, either in the diet or in the gut, does not dictate their functional impact on their host's nutrition or development.

## The ability of bacteria to compensate nutritional deficiencies does not always correlate with the ability of bacteria to synthesize the nutrient

Next, based on the genome-based predictions and the experimentally revealed auxotrophies of GF larvae on FLYAA HD, we correlated the ability of each bacterium to synthesize a nutrient to its ability to fulfill the larval requirements in this nutrient. We identified 4 distinct situations related to the 19 compensations of the 22 auxotrophies shown by GF larvae.

Situation 1: the bacteria synthesize the missing nutrient in the diet and compensate for the related larval auxotrophy (15/19 auxotrophy compensations). In most of the tested conditions, when the bacteria can synthesize a nutrient, they can also fulfill the related nutritional requirements of the GF larvae. For Ap<sup>WJL</sup>, this includes all EAAs<sup>Fly</sup>, Asn, and most vitamins (except pantothenate). For Lp<sup>NC8</sup>, the correlation between the nutritional complementation of ex-GF larva and the ability of Lp<sup>NC8</sup> to synthesize the missing nutrient is more limited and only applies to the requirements of His, Lys, Met, Thr, Asn, and most vitamins. Nonetheless, these results suggest that bacteria can actively supply the nutrients lacking in the HD to the larvae. This phenomenon is reminiscent of previous observations using conventional and gnotobiotic hosts, in which microbial provision of riboflavin or thiamine by host-associated bacteria have been proposed [33,80]. Exceptions to this case seem to be Ap<sup>WJL</sup> on HDΔPantothenate and Lp<sup>NC8</sup> on HDΔTrp. Specifically, Ap<sup>WJL</sup> can produce pantothenate and grows in HDΔPantothenate, and similarly, Lp<sup>NC8</sup> can produce Trp and grows in HDΔTrp. However, neither supported larval development on the respective depleted HD. It is therefore probable that Ap<sup>WJL</sup> and Lp<sup>NC8</sup> produce enough pantothenate and Trp, respectively, to sustain their own growth in the depleted HD, but not sufficiently or in a manner inaccessible to the larvae, and thus fail to fulfill larval requirements for these nutrients.

Situation 2: the bacteria do not synthesize a nutrient, and they cannot fulfill larval nutrient requirements. Expectedly, we observed that when bacteria do not synthesize a nutrient, they

do not fulfill ex-GF larvae requirements for this nutrient. For instance, Lp$^{NC8}$ cannot produce the BCAAs (Ile, Leu, and Val) nor grow in their absence, and thus, it cannot fulfill larval requirements for these amino acids. In some depleted diets, bacteria were able to grow (Fig 3A) even though they cannot synthesize the missing nutrient (Fig 1, S1 and S2 Tables), and they failed to fulfill the larvae requirements of these specific nutrients. This is observed for Ap$^{WJL}$ and Lp$^{NC8}$ on HDΔCholesterol. The likely explanation is that cholesterol is an animal sterol but is dispensable for bacterial growth [67,81]. Similarly, on HDΔCholine and HDΔPyridoxine, Lp$^{NC8}$ grows (Fig 3A) but is unable to fulfill larval requirements (Fig 4A) because according to genome-based predictions, it cannot synthesize these compounds (Fig 1 and S2 Table).

Situation 3: the bacteria do not synthesize a nutrient, but they can fulfill larval nutrient requirements (3/19 auxotrophy compensations). In most cases, we observe growth rescue by bacteria provision of the missing nutrients, but there are interesting exceptions. According to genome-based predictions, Ap$^{WJL}$ is unable to synthesize de novo choline, pyridoxine, and nicotinate (Fig 1B and S2 Table). Surprisingly, it compensates larval auxotrophies on HDΔCholine, HDΔPyridoxine, and HDΔNicotinate. Similarly, genome analysis predict that Lp$^{NC8}$ cannot synthesize nicotinate (Fig 1B and S2 Table), but it compensates larval auxotrophy on HDΔNicotinate.

To confirm that the bacteria are uncapable to synthesize these compounds, we assessed the presence of these compounds in bacterial supernatants using Nuclear Magnetic Resonance (NMR) spectroscopy and High-Performance Liquid Chromatography coupled with Mass Spectrometry (HPLC-MS). We were able to quantify choline in a complete HD using NMR spectroscopy (at 0.531 ± 0.003 mM for a theoretical concentration of 0.477 mM). However, we failed to detect it in the supernatant of Ap$^{WJL}$ culture in HDΔCholine (see Materials and methods section). Similarly, we did not detect any production of nicotinate by either Ap$^{WJL}$ and Lp$^{NC8}$ or production of pyridoxine by Ap$^{WJL}$ in HDΔNicotinate or HDΔPyridoxine, although the analytical method used (HPLC-MS; see Materials and methods) was very sensitive, with a limit of detection of 15.625 nM and 0.977 nM for nicotinate and pyridoxine, respectively.

In the case of choline, Ap$^{WJL}$ may synthesize other compounds that *Drosophila* can use to functionally replace choline. As stated before, *Acetobacter* mutants precluding PC synthesis shift their membrane composition towards increased content of PE and PG [66]. PE and PG have been reported to be part of the phospholipidic repertoire of *Drosophila* membranes [82], in which PE represents approximately 50% of their lipid composition [83]. We posit that ex-GF larvae growing on HDΔCholine capitalize on ethanolamine or glycerol phosphoderivatives produced by Ap$^{WJL}$ to compensate for the lack of choline in their diet.

In the case of pyridoxine, despite its inability to synthesize pyridoxine, Ap$^{WJL}$ may fulfill larval requirements through the production of intermediates such as pyridoxine phosphate, pyridoxal-5-phosphate, or pyridoxamine, which are predicted to be synthesized based on genome analysis (Fig 1B and S2 Table).

Regarding nicotinate, both Ap$^{WJL}$ and Lp$^{NC8}$ grow on HDΔNicotinate and can also fulfill the larval requirements in this vitamin, even though they cannot synthesize it. However, genome-based metabolic predictions suggest that Ap$^{WJL}$ may compensate for the lack of nicotinate by producing intermediates such as nicotinate-D-ribonucleotide, NAD$^+$, and NADP$^+$. In the case of Lp$^{NC8}$, we postulate the existence of alternative metabolic pathways leading to nicotinate intermediate biosynthesis.

Lp$^{NC8}$ cannot grow in the absence of Phe (Fig 3A). The genomic analyses point to the possible loss of the gene coding for the enzyme prephenate dehydratase (4.2.1.51), the penultimate step on Phe biosynthesis, yet Lp$^{NC8}$ can fulfill larval requirements for Phe (Fig 4A). We wondered whether the Phe auxotrophy we observed in 96-well plates (Fig 3A) was due to the oxygenation generated by the agitation through OD readings, as for Thr, Ala, and Asp (Fig 3B).

To test this, we set cultures of Lp$^{NC8}$ in HDΔPhe in static 15-mL closed falcon tubes and assessed bacterial growth after 3 days of culture. In contrast to agitation, Lp$^{NC8}$ grows in HDΔPhe up to $10^6$ CFUs in static conditions (Fig 5B), whereas in the complete media (Fig 3B), Lp$^{NC8}$ grows up to $5 \times 10^8$ CFUs. These results indicate that the rescue of larvae DT by Lp$^{NC8}$ in HDΔPhe is still mediated by bacterial nutrient supply. However, the poor growth of Lp$^{NC8}$ in HDΔPhe suggests the existence of an alternative pathway for Phe biosynthesis in absence of the prephenate dehydratase (Fig 1A). As suggested by Hadadi and colleagues [84], Phe might be produced from L-arogenate using a derivative catalysis through the 2.5.1.47 activity, which is encoded in Lp$^{NC8}$ by the *cysD* gene (nc8_2167) (S2 Table).

A second such interesting case is larval development rescue by Lp$^{NC8}$ on HDΔCys. Lp$^{NC8}$ is an auxotroph for Cys (Fig 3A), even in static conditions (Fig 5B). Lp$^{NC8}$-associated larvae develop faster than GF larvae in HDΔCys, though GF larvae are not auxotrophic for Cys (Fig 4A). This beneficial effect of Lp$^{NC8}$ on ex-GF larvae development on HDΔCys is similar to what is observed on a complete HD (Fig 4A, first row). Therefore, this result probably reflects the basal nutrient-independent growth-promoting effect of Lp$^{NC8}$, which relies on the sensing and signaling of the Lp$^{NC8}$ cell wall by its host (S2 Fig) [23] and requires Lp$^{NC8}$ to be metabolically active (Fig 4A, fifth column). Taken together, our results suggest that Lp$^{NC8}$ is able to grow in HDΔCys only in the presence of *Drosophila* larvae. To test this hypothesis, we assessed Lp$^{NC8}$ growth in solid HDΔCys in the absence and the presence of larvae (Fig 5C). Without larvae, Lp$^{NC8}$ grew one log above the inoculum level (approximately $5 \times 10^5$ CFUs/tube) on solid HDΔCys (Fig 5C, "on diet only"). This minimal growth on solid HDΔCys could be due to the Cys reserves from Lp$^{NC8}$ growth in rich media (De Man, Rogosa, and Sharpe [MRS] medium) prior to inoculation or from contaminants in the agar and cholesterol added to prepare the solid HD. Interestingly, in the presence of larvae in the HDΔCys, Lp$^{NC8}$ CFU counts increased over time, reaching approximately $10^8$ CFUs/tube at day 6 (Fig 5C, "on diet with larvae"). These results indicate that in HDΔCys, larvae support Lp$^{NC8}$ growth, probably by supplying Cys or a precursor/derivative. In turn, Lp$^{NC8}$ sensing and signaling in the host promote larval development and maturation. This observation extends the recent demonstration that *Drosophila* and *L. plantarum* engage in a mutualistic symbiosis, whereby the insect benefits the growth of the bacterium in their shared nutritional environment [30]. Here, we discover that Cys is an additional *Drosophila* symbiotic factor also previously referred to as "bacteria maintenance factor" [30].

Situation 4: Bacterial compensation of minerals and metal deficiencies by concentrating traces or by functional compensation (1/19 auxotrophy compensation). We observed that both Ap$^{WJL}$ and Lp$^{NC8}$ would compensate for Cu, Fe, and Zn deficiencies, but not Mg (Fig 4A, second and third columns). Requirements in Cu and Fe were also fulfilled by HK bacteria (Fig 4A, fourth and fifth columns), although larvae associated with HK bacteria in these conditions developed much slower than larvae associated with living bacteria. This suggests that the inert bacterial inoculum contains traces of Cu and Fe accumulated during the overnight growth in rich medium prior to inactivation and inoculation. These accumulated quantities allowed the larvae to develop when Cu and Fe were not supplied in the HD. Surprisingly, Zn requirements were fulfilled by living bacteria only (Fig 4A). We hypothesize that bacteria may concentrate contaminating traces of these elements in the HD and make them more available to larvae. Alternatively, this could be an interesting case of functional complementation that requires further investigation. Indeed, Zn is an important enzymatic cofactor in the biosynthesis of several metabolites by the larva [85]. In the absence of Zn, GF larvae would not produce these compounds; instead, they could be produced by the bacteria and supplied to the ex-GF larvae similarly to the nutritional complementation we observed above for choline (situation 3). Interestingly, a link between Zn response and the microbiota of *Drosophila* has been described

in previous studies. Expression of the Zn transporter *zip-3* is higher in GF *Drosophila* adults midguts than in their conventionally reared (CR) counterparts [27]. Moreover, the genes encoding metallothioneins B and C (*MtnB* and *MtnC*) are more expressed in flies harboring a microbiota than in GF flies [86]. Metallothioneins are intracellular proteins that can store Zn. Their expression, as well as expression of Zn transporters such as *zip-3*, is regulated by intracellular levels of Zn [87]. Altogether, these results suggest that host-associated bacteria may play an important role in the uptake of metals (especially Zn) by *Drosophila* larvae. This idea is reminiscent of recent reports in *Caenorhabditis elegans*, whereby a bacterium promotes worm development upon Fe scarcity by secreting a scavenging siderophore [88].

## *Drosophila*-associated bacteria provide amino acids essential to larval development

Despite the interesting exceptions detailed above, our data establish that in many cases, *Drosophila*-associated bacteria complement the nutritional requirements of their host by synthesizing and supplying essential nutrients. Bacteria can actively excrete amino acids in their environment when they are produced in excess as intracellular byproducts of metabolic reactions [89]. Moreover, the bacterial cell wall is rich in D-amino acids, and it undergoes an important turnover [90,91]. In certain bacterial species, D-amino acids accumulate in the supernatant during growth and act as a signal to undergo stationary phase [92]. Thus, D-amino acids may also contribute to larval nutrition. Indeed, it has been previously shown that D-amino acids (D-Arg, D-His, D-Lys, D-Met, D-Phe, and D-Val) can support growth of GF larvae probably through the action of amino acid racemases [48]. We thus hypothesized that $Ap^{WJL}$ and $Lp^{NC8}$ could provide amino acids to their host by releasing them in the HD. To directly test this hypothesis, we cultured $Ap^{WJL}$ and $Lp^{NC8}$ in liquid HDs lacking each $EAA^{Fly}$ and quantified the production of the corresponding missing $EAA^{Fly}$. We focused on $EAAs^{Fly}$, whose deficiency could be compensated by bacteria in our DT experiments (Fig 4A). In these assays, $Ap^{WJL}$ was cultured under agitation and $Lp^{NC8}$ cultures were grown in both agitated and static conditions (see Materials and methods). After 3 days, we quantified the amino acid concentration from bacterial supernatants using HPLC. We quantified amino acid production by $Ap^{WJL}$ under agitation while growing in HDΔArg, HDΔHis, HDΔIle, HDΔLeu, HDΔLys, HDΔMet, HDΔPhe, HDΔThr, and HDΔVal and observed accumulation of all missing amino acids except for Lys and Met (Fig 6A). For $Lp^{NC8}$, we analyzed the supernatants of HDs that support $Lp^{NC8}$ growth under agitation (Fig 3A): HDΔHis, HDΔLys, and HDΔMet. We also analyzed supernatants from static conditions, HDΔHis, HDΔLys, HDΔMet, HDΔPhe, and HDΔThr. Surprisingly, from all tested conditions, we only detected His accumulation in the supernatant of $Lp^{NC8}$ grown on HDΔHis under agitation (Fig 6B). We did not detect Lys and Met in $Ap^{WJL}$ culture supernatant or $Lp^{NC8}$ culture under agitation supernatant nor His, Lys, Met, Phe, or Thr in $Lp^{NC8}$ static culture supernatants. However, $Ap^{WJL}$ or $Lp^{NC8}$ can both fulfill larval requirements in an HD lacking these amino acids (Fig 4A). We only analyzed supernatants after 72 h of growth, it is therefore possible that we missed the peak of accumulation of the targeted amino acid, which may have taken place at another time point during the growth phase. Also, $Ap^{WJL}$ and $Lp^{NC8}$ may only secrete precursors or catabolites of those amino acids that we did not target in our analysis. Such amino acid derivatives may also be used by the larvae to compensate for the lack of the cognate amino acids in the diets (such as nicotinate or pyridoxine intermediates; see above). Alternatively, the culture conditions of bacteria on a liquid HD are likely to differ from the conditions encountered in the larval guts, and both $Ap^{WJL}$ and $Lp^{NC8}$ could be receiving cues from the larva itself to produce and/or secrete these nutrients. However, we detected Arg, His, Ile, Leu, Phe, Thr, and Val production by $Ap^{WJL}$ and His

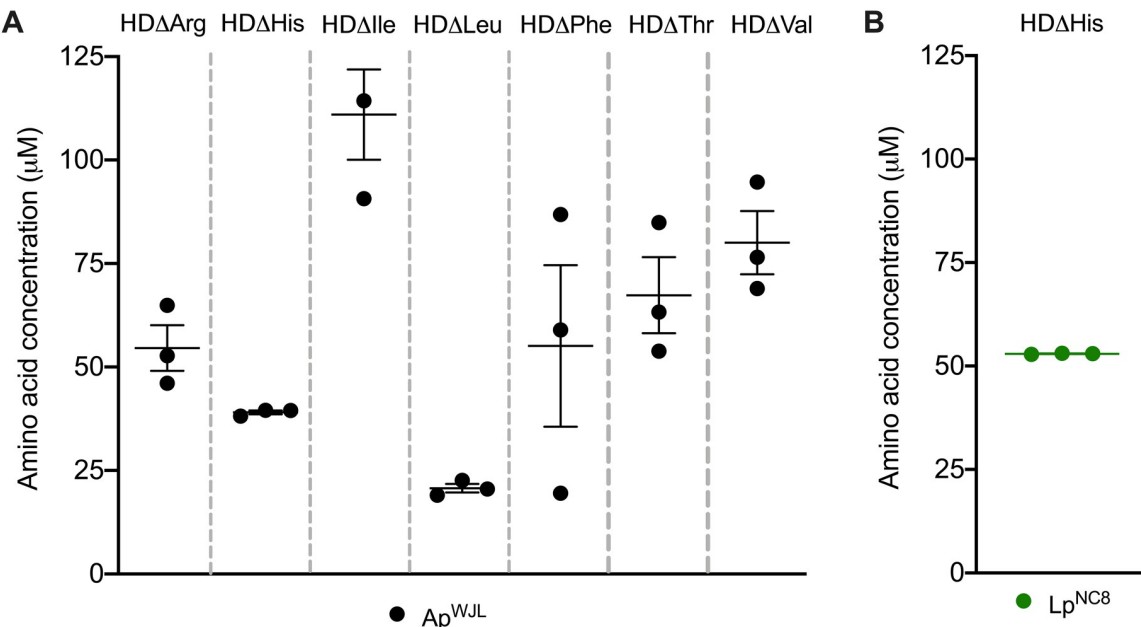

**Fig 6. Ap$^{WJL}$ and Lp$^{NC8}$ can produce and release EAAs$^{Fly}$ during growth.** (A) HPLC measured concentration of Arg, His, Ile, Leu, Phe, Thr, and Val in the supernatant of an Ap$^{WJL}$ culture in HDΔArg, HDΔHis, HDΔIle, HDΔLeu, HDΔPhe, HDΔThr, and HDΔVal, respectively, 72 h after inoculation. Plot shows mean with standard error. Each dot shows an independent replicate. Each amino acid was not detected prior to microbial growth (S1 Data). (B) HPLC measured concentration of His in the supernatant of a Lp$^{NC8}$ culture in HDΔHis, 72 h after inoculation. Plot shows mean with standard error. Each dot shows an independent replicate (53.08 μM, 52.82 μM, and 52.99 μM). Ap$^{WJL}$, *A. pomorum*$^{WJL}$; EAA$^{Fly}$, fly essential amino acid; HD, Holidic Diet; HPLC, High-Performance Liquid Chromatography; Lp$^{NC8}$, *L. plantarum*$^{NC8}$.

by Lp$^{NC8}$, a production that correlates with the respective abilities of Ap$^{WJL}$ and Lp$^{NC8}$ to compensate for the lack of these amino acids in the respective depleted HD. Of note, the concentration of newly synthesized amino acids accumulating in the supernatant is low compared to their concentration in a complete HD (20–150 μM in the former versus 1–5 mM in the latter). However, the bacterial supply of amino acids to the larvae is probably a continuous process, which may also be stimulated upon uptake and transit through the larval intestine. Thus, amino acids would directly be supplied to the larvae and will fulfill its nutritional requirements without the need to accumulate in the surrounding media.

Altogether, our results show that Ap$^{WJL}$ and Lp$^{NC8}$ are able to synthesize and excrete some EAA$^{Fly}$ in their supernatants. These results confirm our hypothesis that *Drosophila*-associated bacteria Ap$^{WJL}$ and Lp$^{NC8}$ produce these EAAs$^{Fly}$ while growing on HDΔEAA$^{Fly}$. When associated with *Drosophila* larvae, Ap$^{WJL}$ and Lp$^{NC8}$ will therefore supply these amino acids to the larvae, allowing larval development on these deficient media as observed upon monoassociations (Fig 4A).

## Conclusion

In this study, we have unraveled the interactions between the nutritional environment of *D. melanogaster* and 2 of its associated bacteria, as well as the functional importance of these interactions for *Drosophila* juvenile growth. We systematically characterized, both in genomes and in vivo, the biosynthetic capacities of growing GF larvae and 2 model bacterial strains behaving as natural partners of *Drosophila* (Ap$^{WJL}$ and Lp$^{NC8}$). We show that both bacteria, each in its unique manner, alleviate the nutritional constraints in the environment to

accelerate host growth and maturation in diets depleted in essential nutrients (Fig 7). The capacity of the bacteria to fulfill 19 of the requirements in 22 essential nutrients for larvae correlated with their metabolic activity and, in most cases (15 out of 19), their ability to produce the missing nutrient. In contrast to obligate symbioses, our results highlight the clear separation between the metabolic pathways of the host and its associated bacteria and reveal a particularly integrated nutritional network between the insect and its facultative bacterial partners around the provision and utilization of nutrients.

Importantly, we further substantiate that the host requirement for essential nutrients can be fulfilled by bacterial provision of a metabolic intermediate of such nutrients (2 out of 19); for example, nicotinate intermediates by both Ap^WJL and Lp^NC8 or pyridoxine intermediates by Ap^WJL. Interestingly, we also detected 2 situations in which nutrient compensation is not explained by a direct supply of the given nutrient or a metabolic intermediate: (i) the compensation of choline deficiency by Ap^WJL and (ii) the compensation of Zn deficiency by both Ap^WJL and Lp^NC8. We propose the existence of functional compensation mechanisms whereby Ap^WJL would complement choline deficiency by synthesizing and providing functional analogues of choline derivatives such as ethanolamine or glycerol derivatives. In addition, both *Drosophila*-associated bacteria would compensate Zn deficiency by uptaking, concentrating, and delivering contaminant traces of Zn to the host.

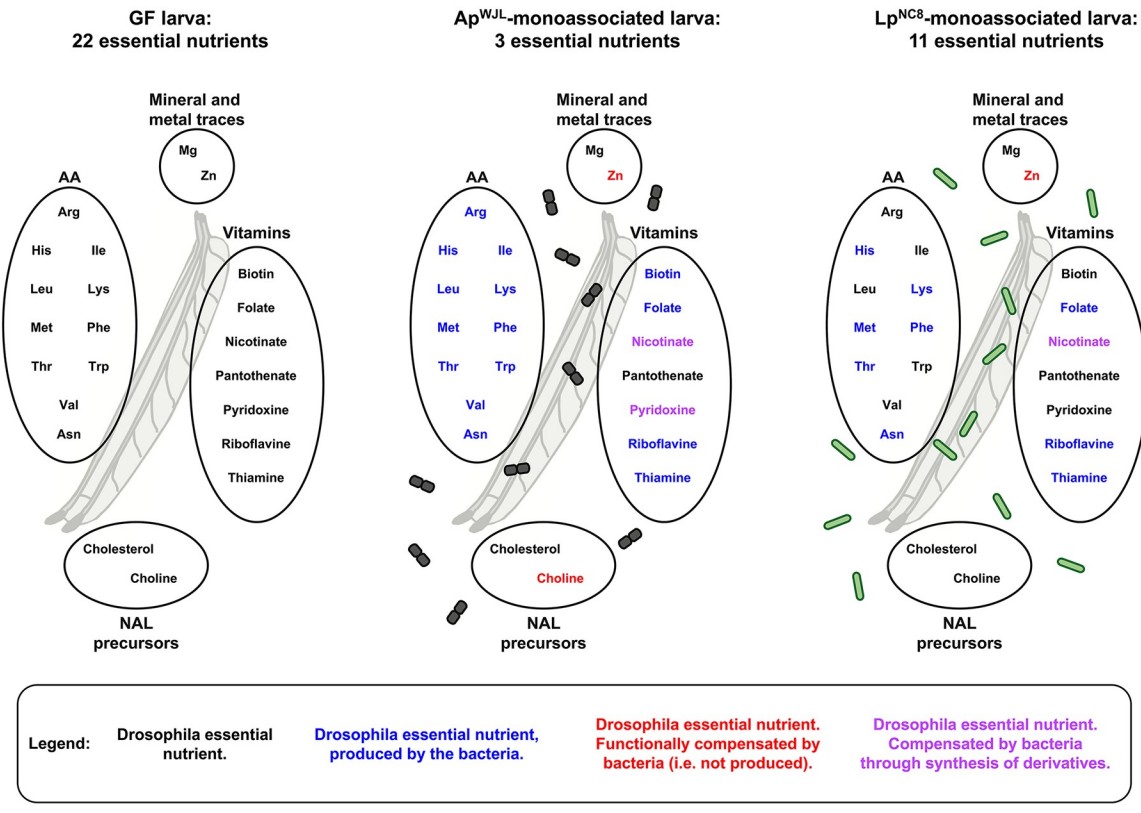

**Fig 7. Ap^WJL and Lp^NC8 differentially shape the nutritional requirements of their juvenile host.** For each gnotobiotic condition, essential nutrients are represented in black and nonessential nutrients in color. Color code: blue, this nutrient can be synthesized by the bacteria; red, this nutrient cannot be synthesized by the bacteria, suggesting a mechanism of functional compensation. In purple: lack of this nutrient may be compensated by an intermediate metabolite or a derivative produced by the bacteria. AA, amino acid; Ap^WJL, *A. pomorum*^WJL; GF, germ-free; Lp^NC8, *L. plantarum*^NC8; NAL, nucleic acid and lipid.

Previous works have shown different mechanisms of growth promotion by microbes in a global low-nutrient context: *Drosophila* larvae can feed on inert microbes to extract nutrients [30,79,93], and living microbes can improve amino acid absorption by increasing the host's intestinal peptidase activity [23,32] and increasing nutrient-sensing–related hormonal signals [20,25]. Here, we show that in addition, the metabolic activities of live *Drosophila*-associated bacteria correct host auxotrophies. These results reveal a novel, to our knowledge, facet of the facultative nutritional mutualism engaged between *Drosophila* and its associated bacteria, which supports the host's nutritional versatility and may allow its juvenile forms to better cope with changes in nutrient availability during the critical phase of postnatal growth, hence ensuring optimal host fitness. Our work lays the basis for further mechanistic studies to investigate whether and how host-associated bacteria regulate the synthesis and release of essential nutrients for the host and whether the host influences this process. Dissecting how bacteria functionally compensate for nutrients that they cannot produce, catabolize excess nutrients, or detoxify toxic molecules also constitutes attractive perspectives for future investigations.

In some cases, the genome-based predictions of bacterial biosynthetic capabilities were incongruent with our in vivo assessment of bacterial auxotrophies (S6 Table). Such seeming discrepancies served as an entry point for us to discover novel, to our knowledge, phenomena and interactions that would have been missed had we only adopted a single approach. One such interesting example is the Asn auxotrophy unique to the *Drosophila yw* line in GF conditions. Another one is the larval provision of Cys (or its derivatives) to Lp^NC8 to maintain a mutualistic nutritional exchange between host and associated bacteria. Previously, a combination of genomic and in vivo approaches has been successfully used for bacteria [62] but not applied to complex symbiotic systems such as facultative host–bacteria nutritional interactions. Indeed, reports characterized these interactions at the genome level [94], but they were not confirmed in vivo. Our work fills this gap and emphasizes the importance of using parallel systematic genome-based annotation, pathway reconstruction, and in vivo approaches for understanding the intricate relationships between the microbial and the nutritional environments and their impact on animal juvenile growth.

## Materials and methods

### Expert automated genome annotation and reconstruction of the biosynthetic potential of *D. melanogaster*, Ap^WJL, and Lp^NC8

We used the CycADS [59], an automated annotation management system, to integrate protein annotations from the complete genomes of *D. melanogaster* (RefSeq GCF_000001215.4 release 6), *A. pomorum* strain DM001 (accession: NCBI Bioproject PRJNA60787), and *L. plantarum* subsp. *plantarum* NC8 (NCBI Bioproject PRJNA67175). CycADS collects protein annotation results from different annotation methods, including KAAS [95], PRIAM [96], Blast2GO [97,98], and InterProScan [99], in order to obtain Enzyme Commission numbers and Gene Ontology annotations. All annotation information was then processed in the CycADS SQL database and automatically extracted to generate appropriate input files to build the 3 BioCyc databases using the Pathway Tools software v22.5 [100]. The BioCyc databases and their associated metabolic networks are available in the EcoCyc database [101]. From the genomic analyses, we inferred the biosynthetic capabilities of the 3 organisms and manually inspected all pathways allowing production of the organic compounds that are present in the exome-based FLYAA HD [40]. For each gap found in biosynthetic pathways or nonconventional enzymatic catalysis, TBLASTN [102] searches were performed in the 3 genomes to look for unpredicted protein activities. Alternative pathways were searched in the literature or using the BioCyc "Metabolic Route Search" tool [103].

## *Drosophila* diets, stocks, and breeding

*D. melanogaster* stocks were reared as described previously [32]. Briefly, flies were kept at 25 ˚C with 12:12-h dark/light cycles on a yeast/cornmeal medium containing 50 g/L of inactivated yeast, 80 g/L of cornmeal, 7.4 g/L of agar, 4 mL/L of propionic acid, and 5.2 g/L of nipagin. GF stocks were established as described previously [86] and maintained in yeast/cornmeal medium supplemented with an antibiotic cocktail composed of kanamycin (50 μg/mL), ampicillin (50 μg/mL), tetracycline (10 μg/mL), and erythromycin (5 μg/mL). Axenicity was tested by plating fly media on nutrient agar plates. *D. melanogaster yw* flies were used as the reference strain in this work. Other *D. melanogaster* lines used include a WT strain from the DGRP collection, DGRP_*25210* [78], and the $w^{1118}$ line [104].

Experiments were performed on HD without preservatives. Complete HD, with a total of 8 g/L of amino acids, was prepared as described by Piper and colleagues using the FLYAAs [40]. Briefly, sucrose, agar, and amino acids with low solubility (Ile, Leu, and Tyr), as well as stock solutions of metal ions and cholesterol, were combined in an autoclavable bottle with milli-Q water up to the desired volume, minus the volume of solutions to be added after autoclaving. After autoclaving at 120 ˚C for 15 min, the solution was allowed to cool down at room temperature to approximately 60 ˚C. Acetic acid buffer and stock solutions for the essential and non-essential amino acids, vitamins, nucleic acids, and lipids were added. Single-nutrient–deficient HD was prepared following the same recipe, excluding the nutrient of interest (named HDΔX, X being the nutrient omitted). Tubes used to pour the HD were sterilized under UV for 20 min. HD was stored at 4 ˚C until use for no longer than 1 week.

## Bacterial strains and growth conditions

$Ap^{WJL}$ [25], $Lp^{NC8}$ [105], $Ap^{WJL}$::Tn*pqq* [25], and $Lp^{NC8}\Delta dlt_{op}$ [23] were used in this study. $Ap^{WJL}$ has been isolated from the midgut of a laboratory-raised adult *Drosophila* [19]. $Lp^{NC8}$ has been isolated from grass silage [105], but we previously showed that it associates effectively with *Drosophila* and benefit its juvenile growth [23]. We use this strain as a model *Drosophila*-associated bacteria thanks to its genetic tractability (no plasmid and high transformation efficiency). *A. pomorum* strains were cultured in 10 mL of Mannitol Broth (Bacto peptone 3 g/L [Becton Dickinson, Sparks, MD USA], yeast extract 5 g/L [Becton Dickinson], D-mannitol 25 g/L [Carl Roth, Karlsruhe, Germany]) in a 50-mL flask at 30 ˚C under 180 rpm during 24 h. *L. plantarum* strains were cultured in 10 mL of MRS broth (Carl Roth) in 15-mL culture tubes at 37 ˚C, without agitation, overnight. Liquid or solid cultures of $Ap^{WJL}$::Tn*pqq* were supplemented with kanamycin (Sigma-Aldrich, Darmstadt, Germany) at a final concentration of 50 μg/mL. CFU counts were performed for all strains on MRS agar (Carl Roth) plated using the Easyspiral automatic plater (Interscience, Saint-Nom-la-Bretèche, France). The MRS agar plates were then incubated for 24–48 h at 30 ˚C for $Ap^{WJL}$ or 37 ˚C for $Lp^{NC8}$. CFU counts were done using the automatic colony counter Scan1200 (Interscience) and its counting software.

## Bacterial growth in liquid HD

To assess bacterial growth in the fly nutritional environment, we developed a liquid HD comprising all HD components except agar and cholesterol. Liquid HD was prepared as described for HD. Single-nutrient–deficient liquid HD was prepared following the same recipe, excluding the nutrient of interest. After growth in culture media, PBS-washed $Ap^{WJL}$ or $Lp^{NC8}$ was inoculated at a final concentration of approximately $10^6$ CFU/mL in 200 μL of either complete liquid HD or nutrient-deficient liquid HD. Cultures were incubated in 96-well microtiter plates (Nunc Edge 2.0; Thermo Fisher Scientific, Waltham, MA, USA) at 30 ˚C for 72 h.

Growth was monitored using an SPECTROstar[Nano] (BMG Labtech GmbH, Ortenberg, Germany) by measuring the optical density at 600 nm ($OD_{600}$) every 30 min. Heatmap in Fig 3A represents the maximal OD detected during the 72 h of growth (average of 3 replicates). The whole experiment was repeated at least twice. Fig 3A was created using the imagesc function in MATLAB (version 2016b; The MathWorks, Natick, MA, USA). $Lp^{NC8}$ growth in static conditions was performed in 10 mL of liquid HD in 15-mL falcon tubes inoculated at a final concentration of approximately $10^4$ CFU/mL. Cultures were incubated at 30 °C for 72 h. After incubation, cultures were diluted in PBS and plated on MRS agar as described above.

## Bacterial growth in solid HD

Bacterial CFUs in HDΔCys were assessed in presence or absence of *Drosophila* larvae. Microtubes containing 400 μL of HD and 0.75- to 1-mm glass microbeads were inoculated with approximately $10^4$ CFUs of $Lp^{NC8}$. Five first-instar larvae, collected from eggs laid on HDΔCys, were added. The tubes were incubated at 30 °C for 0, 3, or 6 days. After incubation, 600 μL of PBS was added directly into the microtubes. Samples were homogenized with the Precellys 24 tissue homogenizer (Bertin Technologies, Montigny-le-Bretonneux, France). Lysate dilutions (in PBS) were plated on MRS, and CFU counts were assessed as described above.

## DT determination

Axenic adults were placed in sterile breeding cages overnight to lay eggs on sterile HD. The HD used to collect embryos always matched the experimental condition. Fresh axenic embryos were collected the next morning and seeded by pools of 40 in tubes containing the HD to test. For the monoassociated conditions, a total of approximately $10^7$ CFUs of $Ap^{WJL}$ or approximately $10^8$ CFUs of $Lp^{NC8}$, washed in PBS, were inoculated on the substrate and the eggs. Inoculation of $Ap^{WJL}$ was limited to approximately $10^7$ CFUs because higher inoculums decreased egg-to-pupa survival. For HK conditions, washed cells of $Ap^{WJL}$ or $Lp^{NC8}$ were incubated for 3 h at 65 °C. Once at room temperature, embryos were inoculated with approximately $10^8$ HK CFUs and approximately $10^9$ HK CFUs, respectively. In the GF conditions, bacterial suspensions were replaced with sterile PBS. Tubes were incubated at 25 °C with 12:12-h dark/light cycles. The emergence of pupae was scored every day until all pupae had emerged. The experiment was stopped when no pupae emerged after 30 days. Each gnotobiotic or nutritional condition was inoculated in 5 replicates. Means, standard error of the mean, and statistical tests (Dunn test of multiple comparisons) are detailed in S4 Table. Because larvae are cannibalistic and can find missing nutrients by eating their siblings [106,107], we therefore excluded replicates with low egg-to-pupa survival (<25%, i.e., $n < 10$). Moreover, we considered that larvae failed to develop in one condition if the mean egg-to-pupa survival of the 5 replicates was inferior to 25% (for details on egg-to-pupae survival, see S5 Table). $D_{50}$ was determined using D50App (http://paulinejoncour.shinyapps.io/D50App) as described previously [23]. The whole experiment was repeated at least twice. $D_{50}$ heatmap represents the average of the 5 replicates of each gnotobiotic and nutritional condition. Fig 4A was done using the imagesc function on MATLAB (version 2016b; The MathWorks).

## Nicotinate and pyridoxine quantification by HPLC/MS

After growth in culture media, PBS-washed $Ap^{WJL}$ or $Lp^{NC8}$ was inoculated in triplicates at a final concentration of approximately $10^6$ CFU/mL into 10 mL of liquid HDΔNicotinateΔPyridoxineΔCholine and HDΔNicotinate, respectively. $Ap^{WJL}$ was grown under agitated conditions (50-mL flasks incubated at 30 °C under 180 rpm). $Lp^{NC8}$ was grown under static conditions (15-mL falcon tubes at 30 °C). Samples were taken at times 0 h and 72 h. Samples

were centrifuged (5,000 rpm, 5 min). Supernatants were collected and stored at −20 ˚C until use. Supernatants were separated on a PFP column (150 × 2.1 mm i.d., particle size 5 μm; Supelco, Bellefonte PA, USA). Solvent A was 0.1% formic acid in H20, and solvent B was 0.1% formic acid in acetonitrile at a flow rate of 250 μL/min. Solvent B was varied as follows: 0 min, 2%; 2 min, 2%; 10 min, 5%; 16 min, 35%; 20 min, 100%; and 24 min, 100%. The column was then equilibrated for 6 min at the initial conditions before the next sample was analyzed. The volume of injection was 5 μL. High-resolution experiments were performed with a Vanquish HPLC system coupled to an Orbitrap Qexactive+ mass spectrometer (Thermo Fisher Scientific) equipped with a heated electrospray ionization probe. MS analyses were performed in positive FTMS mode at a resolution of 70,000 (at 400 m/z) in full-scan mode, with the following source parameters: the capillary temperature was 320 ˚C, the source heater temperature 300 ˚C, the sheath gas flow rate 40 a.u. (arbitrary unit), the auxiliary gas flow rate 10 a.u., the S-Lens RF level 40%, and the source voltage 5 kV. Metabolites were determined by extracting the exact mass with a tolerance of 5 ppm. The limit of detection was determined following the ERACHEM guideline [108]. Nicotinate and pyridoxine standards were mixed at 5 μM and diluted 13 times up to $0.48 \times 10^{-3}$ μM. Each solution was injected 3 times. The limit of detection was determined as LOD = $3 \times s'0$, where $s'0$ is the standard deviation of the intercept.

## Choline quantification by RMN

After growth in culture media, PBS-washed Ap$^{WJL}$ was inoculated in triplicates at a final concentration of approximately $10^6$ CFU/mL into 10 mL of liquid HDΔNicotinateΔPyridoxineΔ-Choline. Ap$^{WJL}$ was then grown under agitated conditions (50-mL flasks incubated at 30 ˚C under 180 rpm). Samples were taken at times 0 h and 72 h. Samples were centrifuged (5,000 rpm, 5 min). Supernatants were collected and stored at −20 ˚C until use. Supernatants were analyzed by 1H 1D NMR on a Bruker Ascend 800 MHz spectrometer (Bruker, Billerica, MA, USA) equipped with a CPCI 5-mm cryoprobe. A volume of 540 μL of supernatant was mixed to 60 μL of Trimethylsillyl Propionic Acid (TSP) 10 mM solution in D2O for spectra calibration. A 1D 1H NMR sequence with water presaturation and a pulse angle of 30˚ and a complete relaxation delay of 7 s was used. An acquisition of 64,000 points was acquired (2 s acquisition time) and processed with 256,000 points.

## DNA extraction and AsnS locus analyses

Genomic DNA from 2 adult *yw* flies was extracted as previously described [109]. Briefly, flies were ground in microtubes containing 0.75- to 1-mm glass microbeads and 500 μL of lysis buffer (Tris-HCl 10 mM, EDTA 1 mM, NaCl 1 mM [pH 8.2]) using the Precellys 24 tissue homogenizer (Bertin Technologies). Then, we added Proteinase K (PureLink Genomic DNA extraction kit; Invitrogen, Carlsbad, CA, USA) at a final concentration of 200 μg/mL and incubated the samples at 56 ˚C under 700 rpm agitation for 1 h. Samples were centrifuged at 10,000 × *g* for 2 min, and we collected the supernatant. *AsnS* coding sequence was amplified by PCR (Q5 Pol High Fidelity M0491S; New England BioLabs, Ipswich, MA, USA) using the primers AsnS_F (CGGGCCGCTTCGTTAAAAA) and AsnS_R (TGGAATTCCTCAGACT TGCCA) with a Veriti Thermal Cycler (Applied BioSystems, Foster City, CA, USA). PCR products were purified using the NucleoSpin Gel and PCR Cleanup kit (Macherey-Nagel, Düren, Germany) following manufacturer's instructions. Sequencing was done by Sanger sequencing (Genewiz, Leipzig, Germany) using the following primers: AsnS_F, AsnS_R, AsnS1 (AGGATTATGGAAAGGATCTTCTGCA), AsnS2 (CTCCGGTCGGATTTGCATCA), AsnS3 (TAATGCCAAAGGGGTCTCGG), and AsnS4 (GTGCGCCAGCTGCATTTATC). The whole coding sequence was then assembled and analyzed using Geneious (version 10.1.3;

Biomatters Ltd., Auckland, New Zealand) by mapping on the reference *D. melanogaster* genome (RefSeq GCF_000001215.4 release 6).

## Amino acid quantification by HPLC

After growth in culture media, PBS-washed $Ap^{WJL}$ or $Lp^{NC8}$ was inoculated in triplicates at a final concentration of approximately $10^6$ CFU/mL into 10 mL of each liquid $HD\Delta EAA^{Fly}$ shown to support their growth and in which they fulfill larval requirements (Figs 3A and 4A). For $Ap^{WJL}$, this includes liquid HDΔArg, HDΔHis, HDΔIle, HDΔLeu, HDΔLys, HDΔMet, HDΔPhe, HDΔThr, and HDΔVal in agitated conditions. For $Lp^{NC8}$, this includes liquid HDΔHis, HDΔLys, and HDΔMet in agitated conditions and liquid HDΔHis, HDΔLys, HDΔMet, HDΔPhe, and HDΔThr in static conditions. For agitated conditions, cultures were done in 50-mL flasks and incubated at 30 ˚C under 180 rpm. Static conditions were performed in 15-mL falcon tubes at 30 ˚C. Samples were taken at times 0 h and 72 h. Samples were centrifuged (5,000 rpm, 5 min). Supernatants were collected and stored at −20 ˚C until use.

Amino acid quantification was performed by HPLC from the supernatants obtained at 0 h and 72 h. Samples were crushed in 320 μl of ultrapure water with a known quantity of norvaline used as the internal standard. Each sample was submitted to a classical protein hydrolysis in sealed glass tubes with Teflon-lined screw caps (6N HCl, 115 ˚C, for 22 h). After air vacuum removal, tubes were purged with nitrogen. All samples were stored at −20 ˚C and then mixed with 50 μL of ultrapure water for amino acid analyses. Amino acid analysis was performed by HPLC (Agilent 1100; Agilent Technologies, Massy, France) with a guard cartridge and a reverse phase C18 column (Zorbax Eclipse-AAA 3.5 μm, 150 × 4.6 mm; Agilent Technologies). Prior to injection, samples were buffered with borate at pH 10.2, and primary or secondary amino acids were derivatized with ortho-phthalaldehyde (OPA) or 9-fluorenylmethyl chloroformate (FMOC), respectively. The derivatization process, at room temperature, was automated using the Agilent 1313A autosampler. Separation was carried out at 40 ˚C, with a flow rate of 2 mL/min, using 40 mM $NaH_2PO_4$ (eluent A [pH 7.8], adjusted with NaOH) as the polar phase and an acetonitrile/methanol/water mixture (45:45:10, v/v/v) as the nonpolar phase (eluent B). A gradient was applied during chromatography, starting with 20% of B and increasing to 80% at the end. Detection was performed by a fluorescence detector set at 340 and 450 nm of excitation and emission wavelengths, respectively (266/305 nm for proline). These conditions do not allow for the detection and quantification of cysteine and tryptophan, so only 18 amino acids were quantified. For this quantification, norvaline was used as the internal standard, and the response factor of each amino acid was determined using a 250 pmol/μl standard mix of amino acids. The software used was the ChemStation for LC 3D Systems (Agilent Technologies).

## Supporting information

**S1 Fig. The Asn auxotrophy of the *yw* line is not due to mutations in the *AsnS* gene.** Pairwise alignment of the AsnS coding region sequenced from *D. melanogaster yw* and the AsnS coding region from *D. melanogaster* reference genome, Bloomington #2057. *yw*, *yellow-white*. (TIF)

**S2 Fig. $Lp^{NC8}\Delta dlt_{op}$, but not $Ap^{WJL}$::Tn*pqq*, shows a loss of function of its intrinsic growth-promoting ability in HD.** $D_{50}$ of GF larvae and larvae associated with $Ap^{WJL}$, $Ap^{WJL}$::Tn*pqq*, $Lp^{NC8}$, and $Lp^{NC8}\Delta dlt_{op}$, reared on complete HD. We performed a Kruskal–Wallis test followed by post hoc Dunn tests to compare each gnotobiotic condition to GF. $^*p$-value < 0.05, $^{****}p$-value < 0.0001. $Ap^{WJL}$, *A. pomorum*$^{WJL}$; *dlt*, XXX; $D_{50}$, day when 50% of larvae

population has entered metamorphosis; GF, germ-free; HD, Holidic Diet; Lp$^{NC8}$, *L. plantarum*$^{NC8}$; ns, nonsignificant; *pqq*, pyrroloquinoline-quinone–dependent; Tn, transposon.
(TIF)

**S3 Fig. Quantity of Ap$^{WJL}$ and Lp$^{NC8}$ on different HDs in presence of larvae.** (A) Bacterial load per larva at day 6 postinoculation. Boxplots show minimum, maximum, and median. Each dot shows an independent replicate. (B) Load of Ap$^{WJL}$ and Lp$^{NC8}$ in solid HD in presence of larvae 3 days and 6 days after inoculation. Plot shows mean with standard error based on 3 replicates by assay. Each dot represents an independent replicate. The dashed line represents the level of inoculation at $t = 0$ h ($10^4$ CFUs per tube). Ap$^{WJL}$, *A. pomorum*$^{WJL}$; CFU, colony-forming unit; HD, Holidic Diet; Lp$^{NC8}$, *L. plantarum*$^{NC8}$.
(TIF)

**S1 Table. Inference from genomic analysis of the biosynthetic capabilities for amino acid production in *D. melanogaster*, Ap$^{WJL}$, and Lp$^{NC8}$.** Ap$^{WJL}$, *A. pomorum*$^{WJL}$; f/i, targeted amino acid biosynthesis is feasible/impossible in a depleted medium; Lp$^{NC8}$, *L. plantarum*$^{NC8}$.
(XLSX)

**S2 Table. Inference from genomic analysis of the biosynthetic capabilities for vitamins production in *D. melanogaster*, Ap$^{WJL}$, and Lp$^{NC8}$.** Ap$^{WJL}$, *A. pomorum*$^{WJL}$; f/i: targeted vitamin biosynthesis is feasible/impossible in a depleted medium; Lp$^{NC8}$, *L. plantarum*$^{NC8}$.
(XLSX)

**S3 Table. OD$_{Max}$ of Ap$^{WJL}$ and Lp$^{NC8}$ grown in 39 HDs.** Mean and SEM of OD$_{Max}$ reached by Ap$^{WJL}$ or Lp$^{NC8}$ grown in complete liquid HD (first line) or liquid HD lacking nutrient X ($\Delta X$, lines below) during 72 h of growth. Ap$^{WJL}$, *A. pomorum*$^{WJL}$; HD, Holidic Diet; Lp$^{NC8}$, *L. plantarum*$^{NC8}$; OD$_{Max}$, maximal optical density; SEM, Standard Error of the Mean.
(XLSX)

**S4 Table. D$_{50}$ of larvae in 40 HDs and 5 gnotobiotic conditions.** Mean and SEM of D$_{50}$ of GF larvae or larvae associated with Ap$^{WJL}$, Lp$^{NC8}$, Ap$^{WJL}_{HK}$, and Lp$^{NC8}_{HK}$. n: number of independent replicates for each condition. For each gnotobiotic condition, we performed a Kruskal–Wallis test followed by post hoc Dunn test to compare each nutritional environment to complete HD. Ap$^{WJL}$, *A. pomorum*$^{WJL}$; D$_{50}$, day when 50% of larvae population has entered metamorphosis; GF, germ-free; HD, Holidic Diet; HK, heat-killed; Lp$^{NC8}$, *L. plantarum*$^{NC8}$; SEM, Standard Error of the Mean.
(XLSX)

**S5 Table. Egg-to-pupa survival in 40 HDs and 5 gnotobiotic conditions.** Mean and SEM of egg-to-pupa survival of GF larvae or larvae associated with Ap$^{WJL}$, Lp$^{NC8}$, Ap$^{WJL}_{HK}$, and Lp$^{NC8}_{HK}$. n: number of independent replicates for each condition. For each gnotobiotic condition, we performed a Kruskal–Wallis test followed by post hoc Dunn test to compare each nutritional environment to complete HD. Ap$^{WJL}$, *A. pomorum*$^{WJL}$; GF, germ-free; HD, Holidic Diet; HK, heat-killed; Lp$^{NC8}$, *L. plantarum*$^{NC8}$; SEM, Standard Error of the Mean.
(XLSX)

**S6 Table. Comparison of genome-based metabolic predictions with in vivo auxotrophies and bacterial complementation of larval nutritional deficiencies.** Can partner A synthesize nutrient X? Prediction from automated annotation and metabolic reconstruction (from Fig 1, S1 and S2 Tables). Can partner A grow in the absence of nutrient X? Auxotrophy observed in vivo (from Fig 3A and 3B). Can bacterial partner A promote larval growth on HD $\Delta X$? In vivo complementation of ex-GF larvae requirements (from Fig 4A), y: yes (green), n: no (red).

Hypothesis to explain contradiction: why the different approaches do not always lead to the same conclusion. GF, germ-free; HD, Holidic Diet; NA, Nonapplicable.
(XLSX)

**S1 Data. All experimental data used to generate graphs of this manuscript.**
(XLSX)

## Acknowledgments

We would like to thank Dali Ma for critical reading and editing of the manuscript and valuable suggestions; members of the Ribeiro and Piper labs for advice on how to effectively implement HD preparations; Edern Cahoreau, Maud Heuillet, and Floriant Bellvert from the Metatoul platform of Genopole Toulouse for NMR and LC/MS analysis; the ArthroTools platform of the SFR Biosciences (UMS3444/US8) for fly equipment and facility; and the Bloomington Stock Center for fly lines.

## Author Contributions

**Conceptualization:** Jessika Consuegra, Théodore Grenier, François Leulier.

**Data curation:** Jessika Consuegra, Théodore Grenier, Patrice Baa-Puyoulet, Nicolas Parisot, Hubert Charles, Federica Calevro.

**Formal analysis:** Jessika Consuegra, Théodore Grenier, Patrice Baa-Puyoulet, Nicolas Parisot, Hubert Charles, Federica Calevro.

**Funding acquisition:** Federica Calevro, François Leulier.

**Investigation:** Jessika Consuegra, Théodore Grenier, Patrice Baa-Puyoulet, Isabelle Rahioui, Houssam Akherraz, Hugo Gervais, Nicolas Parisot, Pedro da Silva, Hubert Charles.

**Methodology:** Jessika Consuegra, Théodore Grenier.

**Project administration:** François Leulier.

**Resources:** Patrice Baa-Puyoulet, Nicolas Parisot, Pedro da Silva, Hubert Charles, Federica Calevro.

**Software:** Patrice Baa-Puyoulet, Nicolas Parisot, Hubert Charles, Federica Calevro.

**Supervision:** Jessika Consuegra, François Leulier.

**Validation:** Jessika Consuegra, Théodore Grenier.

**Visualization:** Jessika Consuegra, Théodore Grenier, Patrice Baa-Puyoulet.

**Writing – original draft:** Jessika Consuegra, Théodore Grenier.

**Writing – review & editing:** Jessika Consuegra, Théodore Grenier, Patrice Baa-Puyoulet, Isabelle Rahioui, Houssam Akherraz, Hugo Gervais, Nicolas Parisot, Pedro da Silva, Hubert Charles, Federica Calevro, François Leulier.

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
