## [Editor Report · Decision Letter 0]

19 Aug 2019

Dear Dr Leulier, 

Thank you for submitting your manuscript entitled "Commensal bacteria differentially shape the nutritional requirements of Drosophila during juvenile growth" for consideration as a Research Article by PLOS Biology.

Your manuscript has now been evaluated by the PLOS Biology editorial staff as well as by an academic editor with relevant expertise and I am writing to let you know that we would like to send your submission out for external peer review.

*Please be aware that, due to the voluntary nature of our reviewers and academic editors, manuscripts may be subject to delays during the holiday season. Thank you for your patience.*

Please re-submit your manuscript within two working days, i.e. by Aug 21 2019 11:59PM.

Kind regards,

Lauren A Richardson, Ph.D

Senior Editor

PLOS Biology

---

## [Decision Letter · Decision Letter 1]

19 Sep 2019

Dear Dr Leulier,

Thank you very much for submitting your manuscript "Commensal bacteria differentially shape the nutritional requirements of Drosophila during juvenile growth" for consideration as a Research Article at PLOS Biology. Your manuscript has been evaluated by the PLOS Biology editors, an Academic Editor with relevant expertise, and by several independent reviewers.

As you will read, the reviewers appreciated many aspects of your work. They do raise some concerns, however, that will need to be addressed in a revision. Of particular note, Rev #4 would like to see addressed to which extent the differences in time of development are due to differences in macronutrients availability in the form of bacterial loads. Therefore Rev #4 requests measuring microbial loads in fly food with and without flies in different food/bacteria combinations where an effect was investigated. The academic editor suggests this to be explicitly addressed and discussed in the text. Measuring the bacterial loads in all the conditions and relate them to time of development would be interesting. However, in may be sufficient to measure Acetobacter and Lactobacillus loads in a subset of these. The Academic Editor also believes that Rev #3's point regarding the discrepancies between predicted and observed auxotrophies/requirements is important to address with further experimental work. S/he also believes that the suggestion for screening to find new genes involved in nicotinate pathway of bacteria is outside the scope of this paper. Lastly, the Academic Editor recommends that you reconsider the use of the term "commensal" in the title and throughout the manuscript, using instead “gut-associated bacteria” or, better, “Drosophila-associated bacteria”.

In light of the reviews (below), we will not be able to accept the current version of the manuscript, but we would welcome resubmission of a much-revised version that takes into account the reviewers' comments. We cannot make any decision about publication until we have seen the revised manuscript and your response to the reviewers' comments. Your revised manuscript is also likely to be sent for further evaluation by the reviewers.

Your revisions should address the specific points made by each reviewer. Please submit a file detailing your responses to the editorial requests and a point-by-point response to all of the reviewers' comments that indicates the changes you have made to the manuscript. In addition to a clean copy of the manuscript, please upload a 'track-changes' version of your manuscript that specifies the edits made. This should be uploaded as a "Related" file type. You should also cite any additional relevant literature that has been published since the original submission and mention any additional citations in your response. 

Before you revise your manuscript, please review the following PLOS policy and formatting requirements checklist PDF: http://journals.plos.org/plosbiology/s/file?id=9411/plos-biology-formatting-checklist.pdf. It is helpful if you format your revision according to our requirements - should your paper subsequently be accepted, this will save time at the acceptance stage.

Please note that as a condition of publication PLOS' data policy (http://journals.plos.org/plosbiology/s/data-availability) requires that you make available all data used to draw the conclusions arrived at in your manuscript. If you have not already done so, you must include any data used in your manuscript either in appropriate repositories, within the body of the manuscript, or as supporting information (N.B. this includes any numerical values that were used to generate graphs, histograms etc.). For an example see here: http://www.plosbiology.org/article/info%3Adoi%2F10.1371%2Fjournal.pbio.1001908#s5.

For manuscripts submitted on or after 1st July 2019, we require the original, uncropped and minimally adjusted images supporting all blot and gel results reported in an article's figures or Supporting Information files. We will require these files before a manuscript can be accepted so please prepare them now, if you have not already uploaded them. Please carefully read our guidelines for how to prepare and upload this data: https://journals.plos.org/plosbiology/s/figures#loc-blot-and-gel-reporting-requirements.

Upon resubmission, the editors will assess your revision and if the editors and Academic Editor feel that the revised manuscript remains appropriate for the journal, we will send the manuscript for re-review. We aim to consult the same Academic Editor and reviewers for revised manuscripts but may consult others if needed.

We expect to receive your revised manuscript within two months. Please email us (plosbiology@plos.org) to discuss this if you have any questions or concerns, or would like to request an extension. At this stage, your manuscript remains formally under active consideration at our journal; please notify us by email if you do not wish to submit a revision and instead wish to pursue publication elsewhere, so that we may end consideration of the manuscript at PLOS Biology.

When you are ready to submit a revised version of your manuscript, please go to https://www.editorialmanager.com/pbiology/ and log in as an Author. Click the link labelled 'Submissions Needing Revision' where you will find your submission record. 

Sincerely,

Lauren A Richardson, Ph.D

Senior Editor

PLOS Biology

Reviews

Reviewer #1: Christen Mirth, signed review

This manuscript by Consuegra and colleagues describes a detailed and thorough set of experiments that define the nutrient requirements of the fruit fly, Drosophila melanogaster, and two of its commensal bacteria, Acetobacter pomorum and Lactobacillus plantarum. They match in silico analyses of the biosynthetic capabilities of each organism with in depth experiments exploring the growth capacities of each bacteria, and of Drosophila with and without bacteria on a chemically defined (holidic) diet. This allows them to explicitly address how the nutrient requirements can be met by bacteria under nutrient stress. 

By sequentially dropping out each of the 38 nutrients in the holidic diet, they first determine the precise nutrient requirements for growth of both bacteria and for egg to pupal development in Drosophila. Next, they found that when they co-culture larvae with either bacteria on these drop-out diets, the bacteria are able to rescue larval development for a limited set of nutrients that are essential to larvae. These commensals differ in their ability to compensate, with A. pomorum able to sustain larval development over a greater number of nutrient-depleted diets. The authors supplement these findings by feeding heat killed bacteria to larvae on the holidic drop-out diets. Heat killed bacteria were unable to rescue developmental timing for any of the fly essential amino acid drop-out diets. This eliminates the potential that the observed effects are due to the fact that the bacteria themselves are being used as food.

Finally, the authors conduct experiments to try to understand how the commensal bacteria supply essential nutrients to their host. They do this by culturing bacteria in the drop-out diets for 18/22 amino acids and assessing whether the bacteria secrete the missing amino acid. Curiously, while bacteria rescue developmental timing in larvae for a range of amino-acid drop-out diets (all 22 amino acids for A. pomorum, 15 amino acids for L. plantarum), they were only able to detect 7 and 1 amino acid in the supernatant for A. pomorum and L. plantarum respectively. This highlights that future work will be key to understanding how bacteria deliver the necessary nutrients to their host.

This work demonstrates in an exquisitely precise manner the relationship between a host, it’s microbes, and the host’s diets. This type of in depth analysis is possible in only a few organisms, but will serve for generating broad perspectives on host/commensal interactions under dietary stress. Given that the microbiome has been linked to growth, fitness, and health of a wide range of animals, this work is sure to be of interest to biologists and health scientists. 

I find this work to be very thorough, and the authors have taken care to use appropriate statistical methods. Further, their story is clear and well-explained. I have only a few comments that I think might help. 

In the results section (lines 369-371), it would be useful to briefly outline the rearing conditions of the larvae (especially because the methods come after the results). I was unsure, until I checked the methods, whether these larvae were reared from egg until pupae on the holidic diets. 

The authors mention that they were unable to detect a number of amino acids in the supernatant when their bacteria were cultured on the holidic diet, and discuss potential reasons why. This was especially notable in L. plantarum, where they were only able to detect the secretion of histidine. I think it would be worth raising the point that the culture conditions of bacteria on holidic diet are likely to differ significantly from the conditions in the larval guts, and that both A. pomorum and L. plantarum could very well be receiving cues from the larva itself to produce/secrete these nutrients.

Minor comments:

Line 68: The abbreviation “Dm” is used infrequently throughout the text. My preference would be to use "Drosophila", to avoid introducing additional acronyms. Also, in some parts you use Drosophila, others D. melanogaster. For consistency, I would pick one.

Line 317: Consider changing “On the contrary” to “In contrast”.

---------------

Reviewer #2: 

The study by Consuegara and Greiner et al. looks at the potential of two major gut commensals of Drosophila melanogaster (Dm) to modulate the nutritional requirements of the host. 

It is well known that the gut microbiota plays an important role in nutrition of the host, in particular in the case of malnutrition and during juvenile growth. However, a systematic experimental study to identify which dietary nutrients can be complemented by a gut symbiont has not been conducted. In the current study, the authors first predict - based on genomic data and in silico metabolic network reconstruction - the biosynthetic capacities of Dm germ-free larvae and the two major gut commensals. Then, they carry out an elegant nutritional screen to experimentally determine which nutrients of a defined larval diet can be complemented by the two gut commensals to sustain larval growth.

To the best of my knowledge, there is no other study that has conducted a similar systemic screen to probe which nutritional requirements of the host can be complemented by gut commensals. The results provide many new insights into the nutritional roles of the two major gut commensals of Dm and show that the metabolism of the host and the gut commensals are highly integrated. Of particular interest is the combination of in silico predictions and experimental data, because this combined approach allowed the authors to discovery interactions that otherwise would not have been identified.

Overall this is an exciting study with many new and relevant insights for the field of microbiome research. The experiments seem to have been designed and carried out with great care and the results are well presented in both text and figures. My only two major points are that the manuscript seems to be very long and the language sometimes more complicated than needed.

Major comments:

1. Language could be simplified. The broad readership of PLoS Biology will have difficulties to fully appreciate the relevance of this paper. In particular, phrasing of the abstract is overly complicated. A striking example is the first sentence. Wouldn’t it be as simple as writing “The interplay between nutrition and the gut microbiome plays a central role in determining juvenile growth in animals”? I would recommend to carefully go through the entire manuscript and remove jargon and unnecessarily complicated phrases. I would also recommend to replace the term ‘microbial environment’ with something else throughout the text, because this term could also refer to the environment of the bacteria instead of the bacterial characteristics of the host environment. Why not use ‘composition of the microbiome’ or simply ‘microbiome’ or ‘host-associated microbiome’?

2. My second major concern is that the manuscript seems to be overly long. It is fine that authors take great care in explaining and discussing their results. However, in particular the first section of the Results/Discussion about the metabolic capabilities of the three partners is lengthy (Line 169-265). Most of it could be moved to the supplementary, and authors could simply keep the last paragraph of each section which summarizes each partners’ capabilities as based on the analysed genomic data. I would also recommend that the authors make an effort to go through the other sections and try to shorten to keep the focus on the most relevant findings.

Minor:

3. I would not use the word semi-essential, but rather consistently use 'conditionally essential'

4. Line 50. Instead of ‘transformed’, ‘modulated’ may be a better term.

5. Line 99: May be this is just me, but I do not understand why the finding that the bacteria-mediated growth benefits are only observed under nutrient scarcity would suggest that the microbiota complements nutrients. Previously, the authors and others have shown that bacteria-mediated growth benefits are due to the induction of peptidases and insulin signalling. Can’t these effects be sufficient to explain the beneficial growth? May be provide a better explanation.

6. Line 146: I disagree that this is a ‘unique’ opportunity. Genomes of many gut symbionts and their dedicated hosts are available allowing similar studies.

7. Fig 3B. ‘Growth x-times inoculum’ is a strange way of plotting the data and labelling the axis. Why not showing absolute number of CFUs, e.g before and after growth? Authors state that there was no difference in growth between conditions, but no statistics have been applied. This should be revised.

8. Fig 4A. Why does this heatmap not go from dark red to white as in the previous figure? A heatmap from red to yellow, to green, and to blue makes little sense, and at least to me it is not intuitive which color corresponds to long or short developing time, especially as ‘no development’ is assigned the ‘color’ white.

9. Line 403. A less cryptic section title would be helpful.

10. Line 444. The title of this section also needs to be edited in my opinion. I would suggest writing ‘Fulfills … requirements’. I don’t think ‘Fulfills … auxotrophies’ is what the authors mean here.

11. Line 573: A more conceptually description of case 4 would be desired. I would also replace the word ‘case’ by ‘scenario’ for all four descriptions. As these are not individual cases (like case studies…), but possible types of nutrient complementation.

12. The discussion of the four different ‘cases’/’scenarios’ is interesting. However, it would be good to know how many of the tested nutrients belong to each of these scenarios. Which one is the dominant type?

13. Line 673. Very long sentence. Why not split in two?

14. Line 686. Instead of ‘foundational blueprint’, why not simply say ‘basis’? These and other complicated words should be removed to keep the text less cryptic. (see my point 1)

15. To colonize germ-free Dm, bacteria were inoculated on the growth substrate and the Dm eggs. To test the effect of heat-killed cells, 10x more heat-killed bacterial cells than live bacterial cells were inoculated. This sounds like a lot. However, it is not clear how much the live bacteria grew after inoculation on the substrate. It may be much more than 10x, and hence the bacteria could simply serve as ‘food’ rather than active symbiotic partners for nutrition. Could the authors please comment on this? Likewise, it would be interesting to know more about the bacterial abundance in the gut? Did both bacteria colonize the gut to the same extent under the different nutrient conditions tested that may have influenced their capacity to complement nutrients.

---------------

Reviewer #3: 

The study by Consuegra and coworkers addresses metabolic crosstalk between Drosophila melanogaster and its two commensal bacteria, Lactobacillus plantarum and Acetobacter pomorum. The authors use in silico metabolic network reconstruction to predict the biosynthetic capacities of Drosophila melanogaster and the two commensals. The authors then systematically test the metabolic dependencies of specific nutrients by utilizing a set of 40 different fully defined diets, lacking one nutrient at a time. In general, the research question is valid and experiments are rigorously conducted.

The data obtained in experiments with holidic diets is in good agreement with the in silico reconstruction: germ free Drosophila melanogaster larvae are auxotrophic for nutrients that cannot be synthesized by it and these auxotrophies are compensated by the commensal, in case it has an intact biosynthetic pathway to synthesize the nutrient. In most cases, bacteria need to be living to compensate for the missing nutrient, although some exceptions are found. Moreover, the study uncovers that Drosophila melanogaster and its facultative commensals have separated metabolic pathways, in contrast to earlier findings on obligate symbiosis. While these findings are not particularly surprising, the systematic nature of this study makes it an important reference within its field. The reconstructed pathways serve as valuable resources for anyone working on metabolism of Drosophila melanogaster or its commensal bacteria. 

More surprisingly, the authors find a small number of cases, where the commensal compensates for the host auxotrophy, despite the lack of a (in silico predicted) biosynthetic pathway for the nutrient. However, these potentially interesting cases remain to be further experimentally explored by the authors. 

Although I certainly see the value of this systematic study for the scientists in this specific field, I do not currently find sufficiently strong novel biological insight to recommend publication in a general readership journal, like PLoS Biology. In my view, the study would be perhaps more suitable for a field-specific journal in its current form. However, further understanding and experimental analysis of the cases, where auxotrophy is compensated without a known biosynthetic pathway, would have the potential to make the conclusions significantly more interesting to a broader audience.

Major comments:

1. The authors state: “In the case of pyridoxine, ApWJL may fulfill larval requirements by either a functional compensation or through pyridoxine biosynthesis by non-canonical pathways (see above). We reach the same conclusion regarding nicotinate.”

The bacteria must be able to synthesize nicotinate somehow. This needs to be resolved prior to publication. Some bacteria are able to synthesize nicotinate from both tryptophan, in addition aspartate (e.g. PMID:14700627). Are you sure both pathways are missing from both commensal strains? If yes, discovery of a new nicotinate biosynthesis pathway would certainly increase the scientific impact of the study. Nicotinate (or its downstream metabolites) should be measured in nicotinate-deficient media to test the existence of biosynthesis. Using genetic screening in the commensals one should be able to identify genes differentially affecting bacterial growth on Nicotinate +/- media. 

2. The authors state: “ApWJL may synthesize other compounds that Drosophila can use to replace choline. As stated before, Acetobacter mutants precluding phosphatidylcholine (PC) synthesis shift their membrane composition towards increased content of phosphatidylethanolamine (PE) and phosphatidylglycerol (PG)… We posit that ex-GF larvae growing on HDDCholine may capitalize on ethanolamine or glycerol produced by ApWJL to compensate for the lack of choline in their diet.”

This is an interesting suggestion that can be tested by lipidomics. Experimental evidence to support this suggestion would significantly improve the study.

3. Several conclusions are made to state that the bacteria are unable to synthesize a metabolite (e.g. ApWJL is unable to synthesize de-novo choline and pyridoxine). These conclusions are only based on the in silico predictions and should be experimentally verified. 

Minor comments:

4. Please specify the genotype of Drosophila strain DGRP-RAL-25210 

5. Please justify, why did you choose to use y, w as reference line to study auxotrophies (white encodes a tryptophan transporter).

---------------

Reviewer #4: 

Consuegra et al use gnotbiotic flies and a chemically defined diet to examine the alleviation of malnutrition by fly gut bacteria. The bacteria convert poor food into more healthy food. While this has been broadly known for quite some time, this study goes into molecular detail as to what specific nutrients are produced and how they influence host health, and to what extent genome annotations are predictive in this case. This is a necessary step towards formulation of precision probiotics, and thus this study provides a template for future work. It puts some real world details to the cartoon view that the microbiome can change our health. I think that is the paper's main strength. We now know from this example how bacteria aid host nutrition based on provision of key nutrients. In some respects this is not novel, but the comprehensive nature of the experiments, using the holidic media for in vitro and in vivo tests, and using the Drosophila model mediates that because it sets up future studies.

Overall I think the study design, approach and data are solid and should be published. I have minor comments, mainly to do with the hype, which I found distracting. It would greatly improve the paper to simply focus on the experiments, which are nice. Significantly shortening the Introduction would accomplish that. The extensive presentation of the genome annotations in the results section could be condensed or put in the supplement. The experimental validation of the auxotrophies is much clearer.

Some additional issues that the authors need to address before publication:

There is an issue of whether the effects seen on fly development are due to auxotrophy or due to an overall lack of sufficient macronutrients like protein, fat, and carbohydrate. Auxotrophy means the flies cannot develop without the nutrient. Changes in rate of development are not due to auxotrophy but often a macronutrient limitation, which can be compensated by the bacterial load. The authors find many cases of such compensation. Work from the Ja lab has shown this explicitly, and it should be addressed in the paper (e.g. Yamada et al 2015; Keebaugh et al 2018; Keebaugh et al 2019). Yet while that is a null hypothesis, there certainly can be more precise compensation, which the authors cover to some extent.

To this point, the only additional experiment I would like to see is a measurement of microbial load in the fly food with vs without flies present for each food/bacteria combination where an effect was investigated. This is to address the compensatory effects of having more food due to bacteria. Figure 4C & D have these measurements for two of the conditions. Please expand to cover all the cases where a bacterium can compensate for a missing nutrient. Please address the extent to which microbial load can compensate for the changes in development rate and cite the appropriate literature. References on malnutrition and aphids could be reduced to make room.

Specifically regarding the aphid literature, while this is intersting and a classic example of an evolved metabolic interaction, the evidence here is not that flies have such an interaction with their bacterial community. I agree it is likely, but the present experiments do not address it. Maybe future experiments where many different strains were compared for their effect on wild fly development could reveal evolved interactions between the fly and specific strains? 

Figure 4: what does 'Niche Load' mean? Please pick another term. For instance, I suggest 'food alone' in place of 'medium' and 'food + larvae' in place of 'niche load'.

line 466: In the heat-killed experiments, 10x the number of cells may not be enough to offset bacterial growth on the food. It's only ~3 doublings. So these heat-killed experiments would only reveal compensation for micronutrient deficiencies, which appears to be the case. The requirement for metabolic activity of the bacteria is thus more likely to be based on the bacteria serving as a source of macronutrients -- so this is malnutrition rather than auxotrophy. Rather than doing more experiments, just rephrase this section to note that the bacteria are probably providing macronutrients in the cases where development is still slow. I realize there are some exceptions to this rule, but they're not addressed in this experiment. 

Figure 5: The negative controls of the media before microbial growth (0h) need to be shown. Otherwise how can we rule out contamination, which is common in chemical reagents? Also, Figure 5B apppears to be misplotted. 

Lp NC8 was isolated from grass silage and therefore is not a fly commensal; it should not be referred to that way. In the text, please note the origin of the strain and the reason it was selected over a true commensal (presumably the genetic tractability). It does not diminish the study but could mislead readers if they do not know the strain's history.

The authors refer to in silico analysis but in fact they just ran an automated annotation pipeline and manually checked results. I found that misleading and was waiting for the in silico experiment as I read the paper, but it never came. Computational biologists refer to computational experiments as in silico when an actual experiment has been performed, such as to predict growth rates in FBA models. This is similar to the way experimentalists do something in vitro or in vivo. The results presented here are not an in silico experiment but rather an automated trip to the library. To get rid of this misleading terminology, please change 'in silico' to 'genome annotation' or something like that. For instance, tables S1 and S2 refer to 'genomic analysis', which sounds accurate.

line 28-29 of Abstract: Please remove this sentence. It sounds like a claim for some mystic feature of the microbiome, but as the authors point out in the paper, there are many likely causes, such as incorrect pathway annotations and provision of precursors rather than final products. 

Line 32-35 of Abstract: Please remove this sentence. This handwaving about a 'novel facet of the facultative nutritional mutualism' sounds grandiose, but the preceeding sentence covers what the study found. These aspects of the writing are distracting from the nice scientific evidence presented in the results. 

Line 151: If a nutrient can be "functionally compensate[d]" for, it means the organism is not an auxotroph for that nutrient. It is misleading to suggest otherwise. Malnutrition and auxotrophy are different. Please rephrase so that readers don't get the false impression that something more complicated is occurring. The delineation of the 4 cases in the results is good. Please just be careful throughout the manuscript to keep auxotrophy and malnutrition separate.

Line 207-210: Seems somewhat irrelevant. Cut.

Line 476: I like this section as it clarifies many of the vagaries in terms of what the microbiome is actually doing.

Line 571: check typos

Line 614: typo:  "...to directly test this..."

Line 623: typo:  "accumulation of"

Line 624: typo:  "except"

Line 668: typo:  "evokes"

In the discussion/conclusion, it would be nice to note to what extent the microbial benefit can be due to catabolism of excess, harmful nutrients as opposed to provision of scarce nutrients. 

Methods: Was gap filling (in the pathways) performed? I did not have time to look up the methods paper referenced. It would be worth mentioning it because hypothesized gaps are examined in the results section.

Line 743: Just a note -- autoclaving sucrose can caramelize it, complicating the chemical composition, and causing batch to batch variation. Best to filter sterilize and add after the autoclaving. But it does not seem to have affected the results here.

---

## [Decision Letter · Decision Letter 2]

10 Feb 2020

Dear Dr Leulier,

Thank you for submitting your revised Research Article entitled "Commensal bacteria differentially shape the nutritional requirements of Drosophila during juvenile growth" for publication in PLOS Biology. I have now obtained advice from three of the original reviewers and have discussed their comments with the Academic Editor. 

As you will read, the reviewers all found your revised manuscript much improved. We will likely publish your study, assuming you are willing to make some final minor edits. In particular, we have two points regarding terminology/word choice. First, and most important, the Academic Editor and Rev #4 are not convinced by your arguments to use the term "commensal", especially as you admit there might be a mutualistic interaction occurring. We reiterate our request to use "Drosophila-associated bacteria" instead. Second, in the abstract and in the text you use the term "sparing". We do not believe the concept of nutrient sparing is widely understood and so recommend either defining this concept or using a more intuitive term.

We expect to receive your revised manuscript within two weeks. Your revisions should address the specific points made by each reviewer. In addition to the remaining revisions and before we will be able to formally accept your manuscript and consider it "in press", we also need to ensure that your article conforms to our guidelines. A member of our team will be in touch shortly with a set of requests. As we can't proceed until these requirements are met, your swift response will help prevent delays to publication.

*Copyediting*

*Published Peer Review History*

*Early Version*

*Submitting Your Revision*

Sincerely,

Lauren A Richardson, Ph.D

Senior Editor

PLOS Biology

DATA POLICY:

Thank you for providing your data. Please ensure that figure legends in your manuscript include information on where the underlying data can be found, and ensure your supplemental data file/s has a legend.

Reviews

Reviewer #2: 

I would like to thank the authors for replying to all my suggestions and edit the manuscript accordingly. I have no further comments.

-------------

Reviewer #3: 

The authors have made a substantial effort to address the issues raised in my original review. The revised manuscript is much improved. I am happy to support publication.

-------------

Reviewer #4: 

The authors have sufficiently addressed all of my experimental concerns and done a nice job of intergating the new results and rewriting the manuscript. I think the paper should be accepted. 

Regarding the use of the term "commensal," I agree with the editor that it should be limited, and "Drosophila-associated bacteria" may be better

Grammar & typos:

Line 44-45: "organism, but insufficiently so under certain..."  "organism but insufficiently under"

Line 596: factors  factor

Line 691: change "in silico" to "genomically" or other word

Line 719: "alive Drosophila's commensal bacteria"  "live commensal bacteria"

Line 728: "catabolize excess of nutrients"  "catabolize excess nutrients"

Line 737: "Previously, combination"  "Previously, a combination"

Line 740: change "in silico"  "genomic" ?

---

## [Editor Report · Decision Letter 3]

4 Mar 2020

Dear Dr Leulier,

On behalf of my colleagues, I am pleased to inform you that we will be delighted to publish your Research Article in PLOS Biology. 

Early Version

PRESS 

Kind regards,

Alice Musson

Publication Assistant, 

PLOS Biology

on behalf of

Hashi Wijayatilake,

Managing Editor

PLOS Biology